# A Theory of Dynamic Benchmarks

**Ali Shirali**
University of California, Berkeley

**Rediet Abebe**
Harvard Society of Fellows

**Moritz Hardt**
Max-Planck Institute for Intelligent Systems, Tübingen

## Abstract

Dynamic benchmarks interweave model fitting and data collection in an attempt to mitigate the limitations of static benchmarks. In contrast to an extensive theoretical and empirical study of the static setting, the dynamic counterpart lags behind due to limited empirical studies and no apparent theoretical foundation to date. Responding to this deficit, we initiate a theoretical study of dynamic benchmarking. We examine two realizations, one capturing current practice and the other modeling more complex settings. In the first model, where data collection and model fitting alternate sequentially, we prove that model performance improves initially but can stall after only three rounds. Label noise arising from, for instance, annotator disagreement leads to even stronger negative results. Our second model generalizes the first to the case where data collection and model fitting have a hierarchical dependency structure. We show that this design guarantees strictly more progress than the first, albeit at a significant increase in complexity. We support our theoretical analysis by simulating dynamic benchmarks on two popular datasets. These results illuminate the benefits and practical limitations of dynamic benchmarking, providing both a theoretical foundation and a causal explanation for observed bottlenecks in empirical work.

## 1 Introduction

In response to concerns around the limitations of static datasets as benchmarks, researchers have proposed dynamic benchmarking—a setting where data collection and model building happen iteratively in tandem—as an alternative (Nie et al., 2020; Potts et al., 2021; Kiela et al., 2021; Ma et al., 2021; Gehrmann et al., 2021). In dynamic benchmarking, model builders fit models against the current dataset, while annotators contribute new data points selected to challenge previously built models. In doing so, the hope is that the iterative process results in a more diverse set of test cases that can help induce better model performance.

Though proponents argue "dynamic adversarial data collection, where annotators craft examples that challenge continually improving models, holds promise as an approach for generating such diverse training sets," there is also a recognition that "the long-term benefits or drawbacks of adopting it as a core dataset creation paradigm remain poorly understood." (Wallace et al., 2022) A major concern is that adversarial data collection proliferates idiosyncratic examples that do well in fooling models but eliminate coverage of necessary yet easier test cases. This can, in turn, reduce dataset diversity and limit external validity (Bowman & Dahl, 2021).

A growing line of theoretical and empirical research on static benchmarks has improved our understanding of the strengths and limitations of this setting. In contrast, similar research on dynamic benchmarks has been limited. The high complexity and cost of studying live benchmarks impede experimental work. A stronger theoretical foundation for dynamic benchmarks could help guide empirical explorations of the vast design space, avoiding costly trial-and-error experiments. However

there has been no apparent theory of dynamic benchmarking that could offer provable guarantees about their performance and clarify how issues such as label noise interact with this setting.

## 1.1 OUR CONTRIBUTIONS

In this work, we initiate a theoretical study of dynamic benchmarks. We contribute a versatile formal model of dynamic benchmarks that serve as the basis for our investigation. We start with a fundamental question:

**Question 1:** *Can we design dynamic benchmarks in such a way that models continue to improve as the number of rounds of data collection grows?*

We start from a theoretical model capturing existing implementations of dynamic benchmarking. This model proceeds in multiple rounds interweaving data collection and model building sequentially. In round $t$, model builders face a distribution $\mathcal{D}_t$ and are tasked with finding a classifier $h_t$ that performs well on $\mathcal{D}_t$. We assume that model fitting succeeds in minimizing risk up to a positive classification error $\epsilon > 0$. We further assume that annotators succeed in identifying the failure cases of the current model $h_t$, giving us access to the uniform distribution $\overline{\mathcal{D}}_t$ over the error cases of the model $h_t$. We determine the new distribution $\mathcal{D}_{t+1}$ by mixing $\overline{\mathcal{D}}_t$ and $\mathcal{D}_t$ in some proportion.

We assume a starting distribution $\mathcal{D}_0$ on the instances of interest. We can think of $\mathcal{D}_0$ as the distribution corresponding to standard data collection. Mirroring the motivation for dynamic benchmarking, this distribution might assign little to no weight to important families of instances. In particular, an error set of measure $\epsilon$, which we assume we can achieve from the get-go, might contain many relevant instances. The goal is therefore to converge to well below the $\epsilon$-error level guaranteed by the above assumption. We assume the distribution admits a perfect classifier so that process could, in principle, converge to $0$ error.

In this setting, we show that three rounds are guaranteed to converge to $O(\epsilon^2)$ error. Unfortunately, this is where it ends. In general, there is no reason to expect this dynamic benchmark to progress below $\Omega(\epsilon^2)$ error. Put differently, there is no provable benefit to dynamic data collection beyond three rounds. The cause of our negative result is a form of catastrophic forgetting that mirrors the concerns quoted earlier. As the benchmark moves beyond three rounds, there is provably no way to retain knowledge of instances correctly classified at earlier stages.

Furthermore, we show through experiments that this lower bound may also be encountered in practice, preventing dynamic benchmarks from progressing beyond a small number of rounds. In doing so, we propose a concrete way to simulate the performance of a dynamic benchmark that may be of independent interest in the empirical study of dynamic benchmarks.

There is yet another impediment to successful dynamic benchmarking: Above we considered the case where the underlying learning problem is *realizable*, meaning that there exists a model that achieves $0$ error on the distribution. In practice, *unrealizable* settings where we have label noise are commonplace. Unrealizability can result from, for instance, annotator disagreement where there is an emerging line of work aiming to understand their impact on data diversity, label noise, and model performance. We show that in this unrealizable setting, dynamic benchmarks concentrate on mislabeled instances, losing their representativeness of the underlying distribution.

Though pessimistic, the above negative results may be inherent to the simple sequential design of dynamic benchmarks currently used in practice. To further probe this issue, we ask:

**Question 2:** *Are there more sophisticated dynamic benchmark designs that can guarantee convergence below the error barrier of the standard setting?*

We answer this question in the affirmative by considering a hierarchical model, which recursively uses the above sequential setting as a building block. In this setting, the organizer of a benchmark creates multiple instances of dynamic data collection and combines the outcomes in a particular way, e.g., by ensembling the resulting models and feeding the output into a new instance. We study the setting where the hierarchy has depth two and show that this setting guarantees convergence to error $O(\epsilon^3)$, providing a strict separation with the standard model. Despite the improved performance, this depth-two setting significantly complicates the benchmarking process, and executing a design with further depth may be prohibitive in practice.

In search of alternative designs that can consistently improve the model's performance, we study a complementary design to dynamic benchmarks in Section A of Appendix. Instead of accumulating adversarial examples in a dynamic benchmark, here the model-in-the-loop carries the information by directly using new examples and becoming more complex throughout the process. We also make a natural connection to boosting methods. Despite achieving zero risk theoretically, this alternative has limited applicability due to either slow convergence or computational infeasibility.

In sum, our results indicate that current bottlenecks observed in empirical settings under the sequential model are inherent to the set-up. This can alert practitioners to the limitations of current practice before many rounds of data are collected. Further, more complex designs such as the hierarchical setting can result in improved performance but may suffer from the organizational complexity of data collection. Combined, these results highlight stark tradeoffs in switching from static to dynamic settings and suggest that exploration of the design space for modeling dynamic benchmarks can play an important role.

## 1.2 RELATED WORKS

For an introduction to datasets as benchmarks, see Chapter 8 in Hardt & Recht (2022). Concerns around static benchmarks are summarized in recent works, including adaptivity (Dwork et al., 2015), violation of sample independence in sequentially generated data (Shirali, 2022), and issues of annotator disagreement (Pavlick & Kwiatkowski, 2019; Prabhakaran et al., 2021; Davani et al., 2022).

Numerous new benchmarks and benchmarking systems have recently been proposed that integrate some aspects of dynamic data collection. Adversarial data collection continually adds challenging examples found by annotators for an existing model (Dinan et al., 2019; Nie et al., 2020; Kiela et al., 2021; Potts et al., 2021; Wallace et al., 2022). Empirical studies show this does not necessarily lead to better performance or robustness (Kaushik et al., 2021; Wallace et al., 2022). A dynamic leaderboard periodically renews the test set (Zellers et al., 2021). In response to the fast growth of dynamic benchmarking, various tools and platforms are also developed. For example, platforms for adversarial data collection (Kiela et al., 2021), assistance of annotators to find challenging examples (Bartolo et al., 2021), personalized benchmarking (Narayan et al., 2021), and automatic crowdsourcing of leaderboard submissions (Khashabi et al., 2021). Adversarial filtering, which filters out examples from a static dataset that are identified to be easy for a given model, is another related technique (Paperno et al., 2016; Zellers et al., 2018; Le Bras et al., 2020). Such datasets are susceptible to being biased (Phang et al., 2022) or saturate faster than static datasets (Taori et al., 2020). Le Bras et al. (2020) include theoretical considerations on eliminating bias. The flurry of newly minted benchmarks stands in stark contrast with the scarcity of theory on the topic. Our work was inspired by the thought-provoking discussion of dynamic benchmarks by Bowman & Dahl (2021).

## 2 PROBLEM FORMULATION

Our primary goal in this work is to understand the population-level dynamics that a benchmark design induces. We are centrally interested in capturing what the iterative process of model building and data collection converges to. Consequently, we ignore finite sample issues in our formulation and focus on distributions rather than samples. We assume that model builders successfully minimize risk approximately. While risk minimization may be computationally hard in the worst case, this assumption reflects the empirical reality that machine learning practitioners seem to be able to make consistent progress on fixed benchmarks. While we focus on population-level dynamics in the design of benchmarks, we restrict ourselves to operations with standard finite sample counterparts.

A *dynamic benchmark design* can be represented as a directed acyclic graph where the nodes and edges correspond to classifiers, distributions, and the operations defined below:

1. Model building: Given a distribution, find an approximate risk minimizer.

2. Data collection: Given a model, find a new distribution.

3. Model combination: Combine a set of models into a single model.

4. Data combination: Combine a set of distributions into a single distribution.

$$\mathcal{D}_0 \xrightarrow{\mathcal{A}} h_0 \xrightarrow{H} \overline{\mathcal{D}}_0 \rightarrow \oplus \rightarrow \mathcal{D}_1 \xrightarrow{\mathcal{A}} h_1 \xrightarrow{H} \overline{\mathcal{D}}_1 \rightarrow \oplus \rightarrow \mathcal{D}_2 \xrightarrow{\mathcal{A}} h_2 \rightarrow \cdots$$

Figure 1: Example of a path dynamic benchmark. Symbol $\mathcal{A}$ represents risk minimization, $H$ represents human data collection, and $\oplus$ shows distribution mixing.

We describe these operations in turn. First, we explain model building by defining the notion of risk and risk minimization. The *risk* of a classifier $h\colon \mathcal{X} \to \mathcal{Y}$ on a distribution $\mathcal{P}$ supported on the data universe $\mathcal{X} \times \mathcal{Y}$ with respect to the zero-one loss is defined as

$$R_{\mathcal{P}}(h) = \mathbb{E}_{(x,y)\sim\mathcal{P}} \left[ \mathbb{1}\left\{ h(x) \neq y \right\} \right] \,.$$

An $\epsilon$-*approximate risk minimizer* $\mathcal{A}$ is an algorithm that takes a distribution $\mathcal{P}$ as input and returns a classifier $h\colon \mathcal{X} \to \mathcal{Y}$ such that

$$R_{\mathcal{P}}(h) \leq \min_{h \in \mathcal{H}} R_{\mathcal{P}}(h) + \epsilon \,,$$

where $\mathcal{H}$ is a family of classifiers. In the benchmark setting, risk minimization is a collective effort of numerous participants. The algorithm $\mathcal{A}$ represents these collective model-building efforts.

We abstract data collection as an operation that takes a classifier $h$ and, for an underlying distribution $\mathcal{D}$, returns a new distribution $\overline{\mathcal{D}}$. In the context of adversarial data collection, we assume that annotators can find the conditional distribution over instances on which $h$ errs $\overline{\mathcal{D}} = \mathcal{D}|_{h(x) \neq y}$. This is an idealized representation of data collection, that ignores numerous real-world issues. Nonetheless, this idealized assumption will make our negative results even stronger.

Model combination happens via a weighted majority vote among multiple classifiers. Distribution combination takes a mixture of multiple given distributions according to some proportions.

The final piece in our problem formulation is the ultimate success criterion for a benchmark design. One natural goal of a dynamic benchmark is to output a classifier that minimizes risk on a fixed underlying distribution $\mathcal{D}$. We envision that this distribution represents all instances of interest. There is a subtle aspect to this choice of a success criterion. If we already assume we have an $\epsilon$-approximate risk minimizer on $\mathcal{D}$, why are we not done from the get-go? The reason is that the $\epsilon$-error term might cover instances crucially important for the success of a classifier in real tasks. After all, a 95% accurate classifier can still perform poorly in practice if real-world instances concentrate on a set of small measure in $\mathcal{D}$. The goal is therefore to find classifiers achieving risk significantly below the error level guaranteed by assumption. By asking for successively higher accuracy, we can ensure that the benchmark continues to elicit better-performing models as time goes on.

**Notation.** We define *error set* of a classifier $h$ as $E_h = \{(x,y) \in \mathcal{X} \times \mathcal{Y} \mid h(x) \neq y\}$. Note that $R_{\mathcal{P}}(h) = \Pr_{\mathcal{P}}(E_h)$. We drop $\mathcal{P}$ from $\Pr_{\mathcal{P}}(\cdot)$ when this is clear from the context. We say a classification problem is *realizable* on distribution $\mathcal{P}$ if there is a classifier $f \in \mathcal{H}$ such that $R_{\mathcal{P}}(f) = 0$. Here, $\mathcal{H}$ is the hypothesis class and $f$ is called the *true* classifier. For realizable problems, no uncertainty will be left over $\mathcal{Y}$ when $x \in \mathcal{X}$ is drawn, so we use $\mathcal{P}$ referring to the distribution over $\mathcal{X}$ and define error sets as a subset of $\mathcal{X}$. Let $p_{\mathcal{P}}$ be the probability density function associated with $\mathcal{P}$. We say $\mathcal{P}$ is *conditioned* on $E \subseteq \mathcal{X}$ and denote it by $\mathcal{P}|_E$ if for any $x \in \mathcal{X}$ we have $p_{\mathcal{P}|_E}(x) = p_{\mathcal{P}}(x|E)$. For notational convenience, we sometimes use $\mathcal{P}(x)$ in place of $p_{\mathcal{P}}(x)$. The *support* $\mathrm{supp}(\mathcal{P})$ of a distribution $\mathcal{P}$ is the largest subset of $\mathcal{X}$ such that $\mathcal{P}(x) > 0$ for all $x \in \mathrm{supp}(\mathcal{P})$. Given probability distributions $\mathcal{P}_1, \mathcal{P}_2, \ldots, \mathcal{P}_T$, we denote the mixture distribution with weights $w_t \geq 0$ such that $\sum_t w_t = 1$ by $\mathrm{mix}(\mathcal{P}_1, \mathcal{P}_2, \cdots, \mathcal{P}_T)$, where $p_{\mathrm{mix}}(x) = \sum_{t=1}^T w_t \mathcal{P}_t(x)$. For a set of classifiers $h_1, h_2, \ldots, h_T$, we denote the weighted majority vote classifier by $\mathrm{maj}(h_1, h_2, \cdots, h_T)$.

## 3    PATH DYNAMIC BENCHMARKS

The simplest case of a dynamic benchmark corresponds to the design of a directed path interleaving model building and data collection as illustrated in Figure 1. This is the design most similar to current proposals of dynamic benchmarks and adversarial data collection (Nie et al., 2020; Kiela

et al., 2021). Starting from an initial distribution, at each round, a new classifier is obtained from the latest distribution. The annotators are then asked to find the vulnerabilities of this model. This new insight will be leveraged towards updating the latest distribution. We call this procedure *path dynamic benchmarking*.

---

*Path dynamic benchmarking:* For an underlying distribution $\mathcal{D}$ with true classifier $f$, given initial distribution $\mathcal{D}_0$ and an approximate risk minimizer $\mathcal{A}$, at each round $t$:

1. $h_t = \mathcal{A}(\mathcal{D}_t)$
2. $\overline{\mathcal{D}}_t = \mathcal{D}|_{h_t(x) \neq f(x)}$
3. $\mathcal{D}_{t+1} = \mathrm{mix}(\mathcal{D}_0, \overline{\mathcal{D}}_0, \overline{\mathcal{D}}_1, \overline{\mathcal{D}}_2, \cdots, \overline{\mathcal{D}}_t)$

---

We first formalize the rationale behind path dynamic benchmarks. Ideally, given a perfect, i.e. 0-approximate, risk minimizer, every time the current classifier misclassifies some part of the underlying distribution, annotators reveal that part and the updated classifier will avoid repeated mistakes. Since errors will not be repeated across the sequence, there can be a limited number of very bad classifiers. The following simple lemma formalizes this intuition for a target error level $\alpha > 0$.

**Lemma 3.1.** *For any hypothesis class $\mathcal{H}$, true classifier $f \in \mathcal{H}$, perfect risk minimizer $\mathcal{A}$, underlying distribution $\mathcal{D}$, and initial distribution $\mathcal{D}_0$ such that $\mathrm{supp}(\mathcal{D}_0) \subseteq \mathrm{supp}(\mathcal{D})$, let $(h_t)_{t=0}^{T-1}$ be any sequence of classifiers obtained in a path dynamic benchmark with equally weighted $\mathrm{mix}(\cdot)$. Then, for any $\alpha > 0$, there are at most $\frac{1}{\alpha}$ classifiers of risk more than $\alpha$. In other words, $|\{t < T \mid R_{\mathcal{D}}(h_t) > \alpha\}| \leq \frac{1}{\alpha}$.*

See proof on page 15.

The lemma does not guarantee the latest classifier's risk, but it is straightforward to see a random selection of the classifiers after many rounds are accurate with high probability (see Corollary C.1). A more effective way to construct an accurate classifier from the sequence of classifiers is to take their majority vote. In this case, three rounds of model building suffice to find a perfect classifier.

**Proposition 3.2.** *Under the conditions of Lemma 3.1, let $(h_t)_{t=0}^{T-1}$ be any sequence of classifiers obtained in a path dynamic benchmark with uniform mixture weights. If $T \geq 3$, $R_{\mathcal{D}}\big(\mathrm{maj}(h_0, h_1, \cdots, h_{T-1})\big) = 0$.*

*Proof.* From the proof of Lemma 3.1 we know $E_t \cap \mathrm{supp}(\mathcal{D}_t) = 0$. So, $E_t \cap E_{t'} = 0$ for every $t' < t$. The majority vote of $h_t$s will misclassify $x$ if half or more of $h_t$s misclassify $x$. But no two distinct $h_t$s make a common mistake. So, for three or more classifiers, the majority vote classifier is always correct. □

So far, path dynamic benchmarking seems to be a promising choice when a perfect risk minimizer is available and the problem is realizable. The situation changes significantly when we go to approximate risk minimizers.

We first study a three-round path dynamic benchmark. We then show how results would generalize for an arbitrary number of rounds. Our results apply to the case where the initial and underlying distributions $\mathcal{D}_0$ do not need to be identical to the target distribution $\mathcal{D}$. To measure the distance between distributions with respect to a hypothesis class, we use the following notion.

**Definition 3.3** (See Ben-David et al. (2010))**.** *For a hypothesis class $\mathcal{H}$ and distributions $\mathcal{P}_1$ and $\mathcal{P}_2$, the $\mathcal{H}\Delta\mathcal{H}$-distance between $\mathcal{P}_1$ and $\mathcal{P}_2$ is defined as*

$$d_{\mathcal{H}\Delta\mathcal{H}}(\mathcal{P}_1, \mathcal{P}_2) = \sup_{h,h' \in \mathcal{H}} \left| \mathbb{E}_{x \sim \mathcal{P}_1}[\mathbb{1}\{h(x) \neq h'(x)\}] - \mathbb{E}_{x \sim \mathcal{P}_2}[\mathbb{1}\{h(x) \neq h'(x)\}] \right|. \tag{1}$$

The next theorem discusses how path dynamic benchmarking with three rounds performs in the case of an $\epsilon$-approximate risk minimizer.

**Theorem 3.4.** *For any hypothesis class $\mathcal{H}$, true classifier $f \in \mathcal{H}$, underlying distribution $\mathcal{D}$, initial distribution $\mathcal{D}_0$ with $\mathrm{supp}(\mathcal{D}_0) \subseteq \mathrm{supp}(\mathcal{D})$, and any $\epsilon$-approximate risk minimizer $\mathcal{A}$, let $h_0$, $h_1$, and*

$h_2$ be the three classifiers obtained after three model building rounds in a path dynamic benchmark with uniform mixture weights. Then, the risk of the majority vote classifier is bounded by

$$R_{\mathcal{D}}\big(\mathrm{maj}(h_0, h_1, h_2)\big) \leq O\left(\epsilon^2 + \epsilon d_{\mathcal{H}\Delta\mathcal{H}}(\mathcal{D}_0, \mathcal{D})\right). \tag{2}$$

*Note that for sufficiently similar $\mathcal{D}_0$ and $\mathcal{D}$, i.e., $d_{\mathcal{H}\Delta\mathcal{H}}(\mathcal{D}_0, \mathcal{D}) = O(\epsilon)$, risk is bounded by $O(\epsilon^2)$.*

See proof on page 15.

The obtained $O(\epsilon^2)$ error with only three rounds of model building is a significant improvement to the $O(\epsilon)$ error that could be achieved with static benchmarks and an $\epsilon$-approximate risk minimizer. We then consider what happens if we continue dynamic benchmarking for many rounds.

**Theorem 3.5.** *For any $\epsilon$-approximate risk minimizer $\mathcal{A}$ with $\frac{1}{\epsilon} \in \mathbb{N}$, hypothesis class $\mathcal{H}$ with $\mathrm{VCdim}(\mathcal{H}) \geq \frac{8}{\epsilon^2}$, and any path dynamic benchmark with $L \geq 3$ rounds of model building and arbitrary mixture weights, there exists an underlying distribution $\mathcal{D}$ such that for any true classifier $f \in \mathcal{H}$ and initial distribution $\mathcal{D}_0$ with $\mathrm{supp}(\mathcal{D}_0) \subseteq \mathrm{supp}(\mathcal{D})$, there exists a sequence $(h_t)_{t=0}^{L-1}$ of classifiers consistent with path dynamic benchmark where the risk of their weighted majority vote is lowerbounded by*

$$R_{\mathcal{D}}\big(\mathrm{maj}(h_0, h_1, \cdots, h_{L-1})\big) \geq \frac{\epsilon^2}{8} \tag{3}$$

*for any weighting of $\mathrm{maj}(\cdot)$. Further, Theorem C.2 shows for any path dynamic benchmark, there exists $\mathcal{H}$ with constant VC dimension such that a similar lower-bound holds.*

See proof on page 16.

Theorem 3.5 shows that $\Omega(\epsilon^2)$ error serves as a lower bound in the approximate risk minimizer setting for any path design (any mixture weighting and weighted majority). Then Theorem 3.4 shows three rounds of model building with uniform weights can achieve the lower bound, so it is optimal, and continuing dynamic benchmarking for more rounds might not be helpful.

### 3.1 Challenges in non-realizable settings

For many reasons, our problem may not be realizable. For example, a class of functions that is not complex enough to explain the true model constitutes an unrealizable setting. Even for a complex enough class, annotators might fail to label instances correctly. In a simplified model for unrealizable problems, we assign random labels to a small part of the distribution and let the rest of the distribution be realizable. Formally, let $\mathcal{X}^\delta$ be the randomly labeled subset of $\mathcal{X}$ and $\mathcal{X}^{\overline{\delta}} = \mathcal{X} \setminus \mathcal{X}^\delta$ be the rest of the domain labeled with $f$. Since no classifier can do well on $\mathcal{X}^\delta$, dynamic benchmarks are prone to overrepresent $\mathcal{X}^\delta$. This intuition is formalized in Theorem 3.6 where we show a significant portion of $\mathcal{D}_t$ will be concentrated on $\mathcal{X}^\delta$.

**Theorem 3.6.** *For any hypothesis class $\mathcal{H}$, true classifier $f$, $\epsilon$-approximate risk minimizer $\mathcal{A}$, and any underlying distribution $\mathcal{D}$ such that $\delta$-proportion of $\mathcal{D}$ is labeled randomly and the rest is labeled by $f$, if $\delta > \epsilon$, as long as $t = O(\frac{\delta}{\epsilon})$, at least $\Omega(1)$-proportion of $\mathcal{D}_t$ obtained through a path dynamic benchmark will be concentrated around the unrealizable instances, i.e., $\mathrm{Pr}_{\mathcal{D}_t}(x \in \mathcal{X}^\delta) = \Omega(1)$.*

See proof on page 18.

As a direct consequence, classifiers trained on a path dynamic benchmark lose their sensitivity to realizable instances and might show an unexpectedly bad performance on the rest of the distribution.

## 4 Hierarchical dynamic benchmarks

Thus far, we have observed that given an $\epsilon$-approximate risk minimizer, the smallest achievable risk through path dynamic benchmarks is $\Omega(\epsilon^2)$. This observation evokes the idea of possibly achieving a squared risk by adding a new *layer* to the benchmarking routine, which calls path dynamic benchmarking as a subroutine. Figure 2 shows such a structure with three steps. At each step, path dynamic benchmarks are obtained starting from a mixture of the initial distribution and error distributions of all previous steps. In the second layer, models found in different steps are aggregated. We

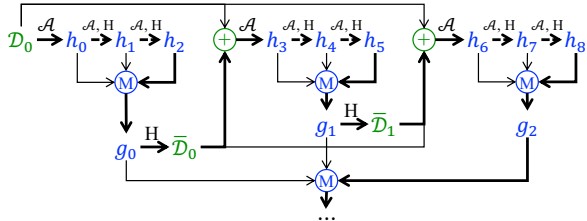

Figure 2: Depth-2 width-3 hierarchical dynamic benchmark. Symbols as in Figure 1 with $M$ representing majority vote.

call the dynamic benchmark of Figure 2 a depth-2 *hierarchical dynamic benchmark*. In an extended form, a depth-$k$ hierarchical dynamic benchmark can be designed by adding a new layer on top of the models obtained from some depth-$(k-1)$ benchmarks.

---

*Hierarchical dynamic benchmarking:* For an underlying distribution $\mathcal{D}$ and true classifier $f$, given an initial distribution $\mathcal{D}_0$ and an approximate risk minimizer $\mathcal{A}$, depth-$k$ width-$w$ hierarchical dynamic benchmarks are constructed recursively:

def. $\mathcal{A}^{(k)}(\mathcal{D}_0)$:

1. $h_0 = \mathcal{A}^{(k-1)}(\mathcal{D}_0)$
2. For $t = 1, \cdots, w-1$:
    (a) $\overline{\mathcal{D}}_{t-1} = \mathcal{D}|_{h_{t-1}(x) \neq f(x)}$
    (b) $\mathcal{D}_t = \mathrm{mix}(\mathcal{D}_0, \overline{\mathcal{D}}_0, \overline{\mathcal{D}}_1, \cdots, \overline{\mathcal{D}}_{t-1})$
    (c) $h_t = \mathcal{A}^{(k-1)}(\mathcal{D}_t)$
3. return $\mathrm{maj}(h_0, h_1, \cdots, h_{w-1})$

where $\mathcal{A}^{(0)} = \mathcal{A}$.

---

Note that this setting does not mean that different steps of path dynamic benchmarking can be done in isolation. Running two benchmarking tasks independently does not yield benefits since humans in the loop might return similar vulnerabilities. In the depth-2 hierarchy of Figure 2, we have bold-faced arrows corresponding to the sequence of steps that should be taken. So, any reasonable dynamic benchmarking based on our model is a sequential process, and the name hierarchy is meant to carry the intuition on how the aggregation of classifiers is happening.

Also note that $\mathcal{A}^{(k)}$ calls annotators for $w^k$ rounds. Since no known empirical dynamic benchmarking study has ever had more than 20 rounds (as studied in Wallace et al. (2022)), we limit our analysis to a depth-2 width-3 structure (Figure 2). Here $w \geq 3$ is the necessary and sufficient number of rounds for path dynamic benchmarks to ensure $O(\epsilon^2)$ risk when $\mathcal{A}$ is $\epsilon$-approximate risk minimizer and $\mathcal{D}_0$ is sufficiently similar to $\mathcal{D}$.

The next theorem provides an upper bound on the risk of any classifier obtained through hierarchical data dynamic benchmarks of Figure 2.

**Theorem 4.1.** *For any hypothesis class $\mathcal{H}$, true classifier $f \in \mathcal{H}$, underlying distribution $\mathcal{D}$, initial distribution $\mathcal{D}_0$ with $\mathrm{supp}(\mathcal{D}_0) \subseteq \mathrm{supp}(\mathcal{D})$, and any $\epsilon$-approximate risk minimizer $\mathcal{A}$, the risk of any classifier obtained from a hierarchical dynamic benchmark with depth-2 and width-3 (Figure 2) where $\mathrm{mix}(\cdot)$ and $\mathrm{maj}(\cdot)$ uniformly weight the inputs, is bounded by*

$$R_\mathcal{D}(\mathrm{maj}(g_0, g_1, g_2)) \leq O\left(\epsilon^3 + \epsilon^2 d_{\mathcal{H}\Delta\mathcal{H}}(\mathcal{D}_0, \mathcal{D})\right). \tag{4}$$

*Note that for sufficiently similar $\mathcal{D}_0$ and $\mathcal{D}$, i.e., $d_{\mathcal{H}\Delta\mathcal{H}}(\mathcal{D}_0, \mathcal{D}) = O(\epsilon)$, risk is bounded by $O(\epsilon^3)$.*

See proof on page 19.

The hope in adding a new layer to dynamic benchmarking was to obtain a squared risk of the previous layer's risk. So, ideally, a depth-2 hierarchical dynamic benchmark could achieve $O(\epsilon^4)$ error.

But this is not the case; next, we show that the upper bound of Theorem 4.1 is tight up to a constant, and the conjecture of squared risk per layer does not hold.

**Theorem 4.2.** *For any $\epsilon$-approximate risk minimizer $\mathcal{A}$ with $\frac{1}{\epsilon} \in \mathbb{N}$, hypothesis class $\mathcal{H}$ with $\mathrm{VCdim}(\mathcal{H}) \geq \frac{2}{\epsilon^3}$, and any hierarchical dynamic benchmark with depth-2 and width-3 (Figure 2) with arbitrary mixture and majority weights, there exists an underlying distribution $\mathcal{D}$ such that for any true classifier $f \in \mathcal{H}$ and initial distribution $\mathcal{D}_0$ with $\mathrm{supp}(\mathcal{D}_0) \subseteq \mathrm{supp}(\mathcal{D})$, there exists classifiers consistent with hierarchical dynamic benchmark for which*

$$R_{\mathcal{D}}\big(\mathrm{maj}(g_0, g_1, g_2)\big) \geq \frac{\epsilon^3}{2}. \tag{5}$$

*Further, Theorem C.3 shows for any hierarchical dynamic benchmark, there exists $\mathcal{H}$ with constant VC dimension such that a similar lower-bound holds.*

See proof on page 20.

Combined, these results show that design structures that are more intricate than the current practice, as captured by the path dynamic benchmark, can yield improved results but also have strong limitations and are challenging to implement in practice.

## 5 EXPERIMENTS

Theorem 3.5, provides an $\Omega(\epsilon^2)$ lower bound on the risk achievable with an $\epsilon$-approximate risk minimizer. The proof of this theorem is constructive, introducing a bad sequence of distributions and classifiers consistent with the path design with $\Theta(\epsilon^2)$ error. But the theorem does not rule out the existence of a good sequence achieving arbitrarily small risk. An important question is how frequently bad sequences appear in practice, retaining the error above zero even after many rounds.

We study this question by simulating path dynamic benchmarks on two popular static benchmarks, CIFAR-10 (Krizhevsky et al., 2009) and SNLI (Bowman et al., 2015). The details of the data and models are reported in Section B.1 of Appendix. Our aim in these experiments is not to obtain a state-of-the-art model. Instead, we want to study the effectiveness of path dynamic benchmarks in a controlled experimental setting with light models. Our simulation design is similar for both datasets:

1. Train a *base classifier* on the whole dataset and fix it for the next steps.
2. Construct a new dataset from samples correctly classified by the base model and define the true and initial distributions as a uniform (point mass) distribution over these samples. Note that in this case empirical risk on samples weighted according to a point mass distribution is equivalent to risk on that distribution. Since the base model correctly labels all selected samples, the problem is also realizable.
3. Draw multiple *rollouts* of path dynamic benchmarks. A rollout from path dynamic benchmark is a sequence of distributions and models obtained by alternatingly training a new classifier on the weighted extracted dataset and up-weighting (down-weighting) the distribution over misclassified (correctly classified) samples according to a mixture rule with uniform weights. Note that the base model is fixed, and the randomness across rollouts solely comes from different initializations of new classifiers and possible randomness in optimization methods.

There are two deviations from our theoretical study in this design: First, we studied binary classification, but both CIFAR-10 and SNLI define multi-label problems. We believe our main arguments hold for multi-label tasks as well. Second, although the problem is realizable, the solution is unlikely to have zero risk as training a new classifier at each round of a rollout typically consists of non-convex optimization. This is addressed in our theoretical framework as $\epsilon$-approximate risk minimization. Our analysis requires a fixed $\epsilon$ across all rounds which approximately holds in practice.

Figure 3a shows results from 100 rollouts of path dynamic benchmarks simulated on CIFAR-10. At each round, the value on the vertical axis shows the risk of the majority vote of all the classifiers obtained in a rollout so far. The solid line is the average of the majority vote's risk of all rollouts, and the shaded area shows the standard deviation. There are a few observations: First, although zero risk is attainable, path dynamic benchmarks fail to reach it. Second, the variance across rollouts is quite consistent at each round. This shows a good rollout keeps being a good, and likewise with a bad

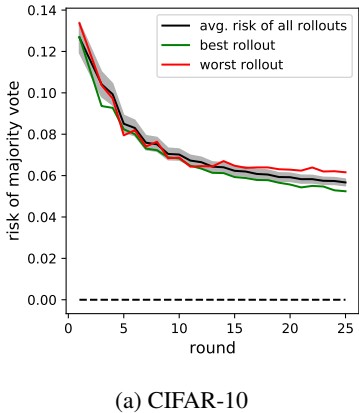
(a) CIFAR-10

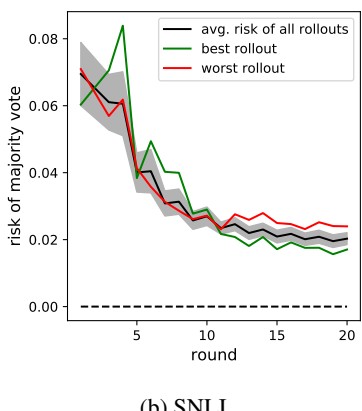
(b) SNLI

Figure 3: Path dynamic benchmark is simulated. The base model is kept fixed for all rollouts. The y-axis shows the risk of the majority vote of all classifiers obtained until that round.

rollout, so continuing dynamic benchmarking does not help. We further discuss why a rollout turns out to be a good or bad one and how this is related to our theoretical negative example in Section B.2 of the Appendix.

Figure 3b shows the results from 50 rollouts of path dynamic benchmarks simulated on SNLI. A similar observation regarding non-zero risk in limit can be made here. Compared to the image classification task, path dynamic benchmarks in the NLI task show more fluctuation through rounds which might be due to the harder nature of the problem or a more complex model.

Summarizing these observations, path dynamic benchmarks, no matter how long we run them for, confront a lower bound even in simple realizable settings. This provides empirical evidence that our negative results are not contrived, but point at what may be inherent limitations to dynamic benchmarking.

# 6 DISCUSSION

The scientific foundation of traditional benchmarks is the holdout method, whereby we split the dataset into a training and testing component. Although the machine learning community routinely operates well outside the statistical guarantees of the holdout method, at least there is some cognizable theoretical framework for static datasets as benchmarks. Moreover, more recent works have found new support for static benchmarks (Blum & Hardt, 2015; Recht et al., 2019).

Dynamic benchmarks provide an intriguing proposal aimed at mitigating known issues with static benchmarks. Platforms for dynamic benchmarks are already up and running. However, machine learning theory has little to offer in the way of guiding the principled design of valid dynamic benchmarks. Responding to this lack of theoretical foundations, we propose a formal model of dynamic benchmarks that allows to combine model building and data collection steps in a flexible manner. We focus on what we think is the first-order concern in the design of a benchmark: Does the benchmark, in principle, induce the creation of better models?

While our results show a provable benefit to dynamic benchmarking, it is arguably more subtle than hoped and comes at a significant increase in complexity. Our negative results are particularly concerning given that we optimistically assume that both model builders and annotators have significant competence. In practice, dynamic benchmarks likely face additional obstacles neglected by our idealized assumptions. As a result, it is less clear to what extent our positive results have prescriptive value. On the other hand, our positive results make it clear that the design space for dynamic benchmarks is larger than currently utilized, thus pointing at a range of interesting open problems. Finally, lowering risk or improving the accuracy of the induced models is the ultimate success criterion of a benchmark, however, there are other concerns including the robustness of the models that can be the topic of future works based on our proposed framework.

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

## A  DYNAMIC GRADIENT-BASED UPDATE

The current practice of dynamic benchmarking aims at diversifying the benchmark by dynamically adding new *hard* examples to it. The hope is to obtain better models from the benchmark every time new samples are added. As we observed, despite the initial benefit this method has, there exist arbitrarily long dynamic benchmarks with no significant improvement after the first few steps.

But if the ultimate goal is to obtain improved models, there is a more direct way to do it. Instead of keeping all the adversarial feedback in the form of a dynamic benchmark, the model-in-the-loop itself can carry all the information by becoming more and more complex throughout the process. In other words, rather than updating the benchmark by adversarial examples, these examples can be collected in a way that can be directly used to update the model. Dynamic adversarial feedback from annotators is helpful here by providing fresh examples at each round that prevent overfitting. This phenomenon is also close to the boosting technique in machine learning.

In this section, we discuss how directly updating the model using adversarial data can be formulated within our framework and why they are not practical. Since we use gradient-based techniques to update the model, we call these methods dynamic gradient-based updating. We first formally introduce a vector notation for functions and distributions, which makes our arguments easier to follow. Then we discuss how a classifier's risk with respect to the zero-one loss would be represented with this notation. In search of a classifier that minimizes this risk, we minimize surrogate convex losses rather than directly optimizing for zero-one risk. Here we discuss two popular choices, to be named hinge and exponential losses, and for each, discuss the corresponding method with its strengths and limitations.

$$\mathcal{D}_0 \xrightarrow{\mathcal{A}} h_0 \xrightarrow{\text{H}} \overline{\mathcal{D}}_0 \xrightarrow{\mathcal{A}} \overline{h}_0 \rightarrow \oplus \rightarrow h_1 \xrightarrow{\text{H}} \overline{\mathcal{D}}_1 \xrightarrow{\mathcal{A}} \overline{h}_1 \rightarrow \oplus \rightarrow h_2 \rightarrow \cdots$$

Figure 4: Dynamic gradient-based update: hinge loss minimization.

**Notation.** Let $h : \mathcal{X} \to \{-1, 1\}$ be a binary classifier defined on the finite set $\mathcal{X}$. The vector representation of $h$ in $\mathcal{X}$ is $\boldsymbol{h} = (h(x))_{x \in \mathcal{X}}$. Similarly, let $\mathcal{P}$ be a probability distribution over $\mathcal{X}$. The vector representation of $\mathcal{P}$ in $\mathcal{X}$ is $\boldsymbol{P} = (\mathcal{P}(x))_{x \in \mathcal{X}}$. The entrywise product of two vectors $\boldsymbol{h}_1$ and $\boldsymbol{h}_2$ is denoted by $\boldsymbol{h}_1 \circ \boldsymbol{h}_2$. For an underlying distribution $\mathcal{D}$ and true classifier $f$, we can use vector representations to write the risk with respect to the zero-one loss. Risk of a binary classifier $h$ on $\mathcal{D}$ is $R_{\mathcal{D}}(h) = \frac{1}{2}\langle \boldsymbol{1} - \boldsymbol{h} \circ \boldsymbol{f}, \boldsymbol{D} \rangle$. For a general $h : \mathcal{X} \to \mathbb{R}$, still we can define the risk with respect to the zero-one loss as $R_{\mathcal{D}}(h) = \frac{1}{2}\langle \boldsymbol{1} - \text{sign}(\boldsymbol{h} \circ \boldsymbol{f}), \boldsymbol{D} \rangle$, where $\text{sign}(\cdot)$ is an entrywise operator.

**Upper-bounded risk.** There are many ways to upper-bound $R_{\mathcal{D}}(h)$. For example, for any entrywise function $l(\cdot)$ such that $l(x) \geq \mathbb{1}\{x \leq 0\} = \frac{1}{2} - \frac{1}{2}\text{sign}(x)$ for all $x \in \mathbb{R}$, the risk of $h$ with respect to the zero-one loss can be upper-bounded by

$$R_{\mathcal{D}}(h) \leq R_{\mathcal{D}}^l(h) = \langle l(\boldsymbol{h} \circ \boldsymbol{f}), \boldsymbol{D} \rangle. \tag{6}$$

### A.1 MINIMIZING HINGE LOSS

A popular function to upper-bound the zero-one risk is the hinge loss: $l(x) = \max(1 - x, 0)$. Plugging $l(\cdot)$ into Equation 6 gives:

$$R_{\mathcal{D}}(h) \leq R_{\mathcal{D}}^{\text{hinge}}(h) = \langle \max(\boldsymbol{1} - \boldsymbol{h} \circ \boldsymbol{f}, \boldsymbol{0}), \boldsymbol{D} \rangle, \tag{7}$$

where $\max(\cdot)$ is element-wise maximization. Let $\boldsymbol{g} = \nabla_h R_{\mathcal{D}}^{\text{hinge}}$. Looking for an update of the form $\boldsymbol{h} := \boldsymbol{h} + \Delta\boldsymbol{h}$ to reduce $R_{\mathcal{D}}^{\text{hinge}}(h)$, any direction such that $\langle \boldsymbol{g}, \Delta\boldsymbol{h} \rangle < 0$ will be a descent direction and a small step size guarantees consistent decrease of $R_{\mathcal{D}}^{\text{hinge}}$. As we will show in the proof of Lemma A.1, directly applying gradient descent, i.e., $\Delta\boldsymbol{h} = -\eta\,\boldsymbol{g}$, is not practical, as it incorporates summation of a distribution and a classifier vector. Unlike classifiers which are known for every point in the domain, in practice, distributions are limited to the available samples and this summation is not implementable. Alternatively, let $E_h = \{x \in \mathcal{X} \mid h(x)f(x) < 1\}$ be the set of margin errors of classifier $h$. We task annotators to return $\overline{\mathcal{D}}_h = \mathcal{D}|_{E_h}$ given $h$. Let $\overline{h} = \mathcal{A}(\overline{\mathcal{D}}_h)$ be the model built on the vulnerabilities of $h$. Next lemma shows $\overline{h}$ is a descent direction for the hinge loss.

**Lemma A.1.** *For any hypothesis class $\mathcal{H}$, true classifier $f \in \mathcal{H}$, current classifier $h \in \mathcal{H}$, $\epsilon$-approximate risk minimizer $\mathcal{A}$, and any underlying distribution $\mathcal{D}$, the vector representation $\overline{\boldsymbol{h}}$ of the classifier $\overline{h} = \mathcal{A}(\mathcal{D}|_{h(x)f(x)<1})$ is a descent direction for $R_{\mathcal{D}}^{\text{hinge}}(h)$.*

See proof on page 22.

This lemma lets us write the updating rule $\boldsymbol{h} := \boldsymbol{h} + \eta\, R_{\mathcal{D}}^{\text{hinge}}(h)\,\overline{\boldsymbol{h}}$, depicted graphically in Figure 4. Since gradient dominance condition holds for this update and hinge loss is 1-Lipschitz, $\boldsymbol{h}$ will converge to $\boldsymbol{f}$ with the rate of $O(\frac{|\mathcal{X}|}{t^2})$. Although this method guarantees convergence, the dependence on the domain size makes the bound useless for continuous or very large domains.

### A.2 MINIMIZING EXPONENTIAL LOSS

Another candidate function to upper-bound the zero-one risk is the exponential loss: $l(x) = \exp(-x)$. This leads to a similar analysis as the AdaBoost algorithm (Schapire & Singer, 1999). Plugging $l(\cdot)$ into Equation 6 gives:

$$R_{\mathcal{D}}(h) \leq R_{\mathcal{D}}^{\exp}(h) = \langle \exp(-\boldsymbol{h} \circ \boldsymbol{f}), \boldsymbol{D} \rangle, \tag{8}$$

where $\exp(\cdot)$ is element-wise exponential function. Similar to the hinge loss minimization, we show in the proof of Lemma A.2 that directly updating $\boldsymbol{h}$ with a gradient term is not implementable. So,

we search in the hypothesis class for a classifier $\tilde{h}$ such that $\tilde{\boldsymbol{h}}$ minimizes $\langle \tilde{\boldsymbol{h}}, \boldsymbol{g} \rangle$, where $\boldsymbol{g} = \nabla_h R_{\mathcal{D}}^{\text{exp}}$. Next lemma finds such a classifier along with the optimal step size.

**Lemma A.2.** *For any hypothesis class* $\mathcal{H}$, *true classifier* $f \in \mathcal{H}$, *current classifier* $h \in \mathcal{H}$, $\epsilon$-*approximate risk minimizer* $\mathcal{A}$, *and any underlying distribution* $\mathcal{D}$, $\tilde{h} = \mathcal{A}(\mathcal{D}_h)$ *is the solution of* $\min_{\tilde{h} \in \mathcal{H}} \langle \tilde{\boldsymbol{h}}, \boldsymbol{g} \rangle$. *Here* $\mathcal{D}_h(x) \propto \mathcal{D}(x) \exp(-h(x)f(x))$ *and* $\boldsymbol{g} = \nabla_h R_{\mathcal{D}}^{\text{exp}}$. *Further,* $\eta = \frac{1}{2} \log(\frac{1}{R_{\mathcal{D}_h}(\tilde{h})} - 1)$ *is the best step size for the update rule* $\boldsymbol{h} := \boldsymbol{h} + \eta \, \tilde{\boldsymbol{h}}$.

See proof on page 23.

Let $h_t$ be the final classifier obtained after $t$ updates according to the updating rule of Lemma A.2. An analysis similar to the analysis of AdaBoost shows $R_{\mathcal{D}}(h_t) \leq \exp(-\frac{(1-2\epsilon)^2 t}{2})$. This method, despite the exponential convergence rate, is not practical for two reasons. First, it is computationally hard as reweighting a distribution requires the calculation of a normalization factor which is a sum over the whole domain. Second, it requires sampling from $\mathcal{D}$ which might not be possible.

In summary, gradient-based updates guarantee convergence of the updated classifier to the true classifier; however, they either suffer from slow convergence or computational hardness.

# B MORE ON EXPERIMENTS

We elaborate on the details of datasets, models, and further observations from the simulation of path dynamic benchmarks in this section.

## B.1 DATA AND MODELS

The CIFAR-10 dataset contains $60,000$ of $32 \times 32$ color images in 10 different classes, commonly used as a static benchmark for image classification. We use a shallow feed-forward Convolutional Neural Network (CNN) consisting of two convolution layers (with 32 and 64 filters), interleaved with two max-pooling layers, followed by a dropout layer and a dense layer with 10 units at the output, increasing total number of trainable parameters to 40k. We use the same architecture to train a base model and train classifiers in drawing rollouts. The base classifier achieves $73\%$ accuracy after 30 epochs on the training data, reasonably above the chance level of about $10\%$.

The Stanford Natural Language Inference (SNLI) corpus (version 1.0) is a popular NLI benchmark consisting of 570k human-written English sentence pairs manually labeled for balanced classification with the labels entailment, contradiction, and neutral. We restrict our simulation to a 50k random subset of this data. Our model is a slightly modified model of Bowman et al. (2015) where words are represented with pre-trained 100d GloVe[1] vectors, and two parallel Bidirectional Long Short-Term Memory layers (BiLSTM) are used to extract sentence embeddings. The concatenated embeddings are then fed into three more hidden dense layers (each has 128 units) before going to the last dense layer with 3 units. The model has a total number of 120k trainable parameters, and the base model achieved an accuracy of $68\%$ on the training data and $61\%$ on the test data, consistent with the original study when the number of samples was limited.

## B.2 FURTHER OBSERVATIONS

The observations we had in Section 5 raise another question: what makes a rollout a bad rollout? Do bad rollouts share a common characteristic? We hypothesize the more similar a rollout to the bad sequence constructed in the proof of Theorem 3.5, the worse its final risk. A unique feature of the negative example in Theorem 3.5 is that initial classifiers in the sequence cleverly choose their error sets to only overlap on a fixed part of the distribution, which will turn out to be the error set of the majority. We define a score that captures this behavior without being directly related to the accuracy of the final majority vote model. Let $E_m$ be the error set for the majority vote classifier and $E_{h_t}$ be the error set of the round $t$ classifier. Then for every pair of distinct rounds $t_1, t_2 < T$ of a rollout, define $z_{t_1, t_2} = \frac{|E_{h_{t_1}} \cap E_{h_{t_2}} \cap E_m|}{|E_{h_{t_1}} \cap E_{h_{t_2}}|}$. We use $z_T = \frac{1}{T(T-1)} \sum_{t_1=0}^{T-1} \sum_{t_2=0, t_2 \neq t_1}^{T-1} z_{t_1, t_2}$ to

---

[1] http://nlp.stanford.edu/data/glove.6B.zip

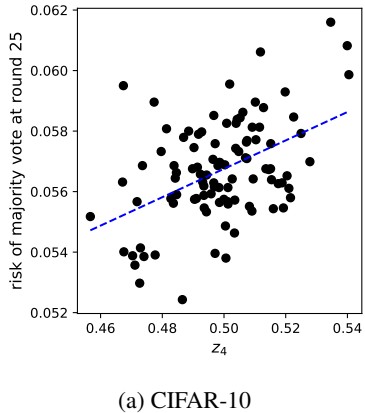
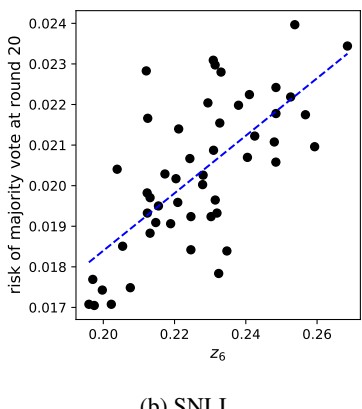

(a) CIFAR-10                                    (b) SNLI

Figure 5: Similarity to the theoretical negative example measured by $z_T$ can partially explain the risk of the majority vote classifier of all rounds. The blue dashed line is the minimum squared error fit.

quantify similarity of a rollout to the theoretical negative example. The proof of Theorem 3.5 shows that as long as $T \leq \frac{2}{\epsilon}$, the theoretical negative example gives $z_T = 1$.

Figure 5a depicts the risk of the majority vote at the end of the rollout (round 25) vs. $z_4$ score. The correlation is positive and statistically significant (Pearson $r = 0.46$, $p < 0.001$). So, the more similar a rollout to the theoretical bad example in terms of the defined score, the more likely it will be a bad rollout. Interestingly, at round 4, the risk of the majority vote classifier can barely explain the risk after 25 rounds (Pearson $r = 0.18$, $p > 0.05$). This shows our negative example construction has not only introduced a worst-case example but might have identified an important characteristic that naturally appears in dynamic benchmarks and limits their effectiveness.

Figure 5b also shows a significant positive correlation (Pearson $r = 0.68$, $p < 0.001$) between the final risk of a rollout and its similarity to the theoretical negative example measured early at round 6. At this round, the risk of the current majority vote classifier is barely informative about the risk after 20 rounds (Pearson $r = -0.27$, $p > 0.05$). Once more, it shows the negative example constructed theoretically can explain the natural failure of dynamic benchmarking in different contexts.

## C  ADDITIONAL STATEMENTS AND PROOFS

**Corollary C.1.** *Under conditions of Lemma 3.1, let $\hat{h}$ be one of the $h_t$s selected uniformly at random. For any $\delta > 0$ and $\alpha > 0$, if $T \geq \frac{1}{\delta\alpha}$, with probability at least $1 - \delta$, $\hat{h}$ is $\alpha$-accurate.*

*Proof.* From Lemma 3.1, the probability that $\hat{h}$ is $\alpha$-bad is less than $\frac{1/\alpha}{T} \leq \delta$.  □

*Proof of Lemma 3.1.* As $\mathrm{supp}(\mathcal{D}_0) \subseteq \mathrm{supp}(\mathcal{D})$, running the path dynamic benchmarking will preserve $\mathrm{supp}(\mathcal{D}_t) \subseteq \mathrm{supp}(\mathcal{D})$ for any $t$. On the other hand, we know $R_{\mathcal{D}}(f) = 0$. So, we have $R_{\mathcal{D}_t}(f) = 0$. As $\mathcal{A}$ is a perfect risk minimizer and $f \in \mathcal{H}$, we should have $R_{\mathcal{D}_t}(h_t) = 0$. Therefore, errors of $h_t$ should happen outside of $\mathrm{supp}(\mathcal{D}_t)$:

$$E_t \cap \mathrm{supp}(\mathcal{D}_t) = 0, \tag{9}$$

where $E_t$ is the error set of $h_t$. Now assume $t$ is one of the indices where $h_t$ is $\alpha$-bad, i.e., $\mathrm{Pr}_{\mathcal{D}}(E_t) > \alpha$. Then we have

$$\Pr_{\mathcal{D}}(x \in \mathrm{supp}(\mathcal{D}_{t+1})) = \Pr_{\mathcal{D}}(x \in \mathrm{supp}(\mathcal{D}_t)) + \Pr_{\mathcal{D}}(x \in E_t) \geq \Pr_{\mathcal{D}}(x \in \mathrm{supp}(\mathcal{D}_t)) + \alpha. \tag{10}$$

This cannot happen more than $\frac{1}{\alpha}$ times.  □

*Proof of Theorem 3.4.* Starting from $\mathcal{D}_0$ where $\text{supp}(\mathcal{D}_0) \subseteq \text{supp}(\mathcal{D})$, path dynamic benchmarks will preserve $\text{supp}(\mathcal{D}_t) \subseteq \text{supp}(\mathcal{D})$ for any $t$. Since the problem is realizable and $\mathcal{A}$ is $\epsilon$-approximate risk minimizer, for any consistent $h_t$ we should have $R_{\mathcal{D}_t}(h_t) \leq \epsilon$. The distribution $\mathcal{D}_t$ itself is a mixture of the initial and error distributions: $\mathcal{D}_t = \text{mix}(\mathcal{D}_0, \overline{\mathcal{D}}_0, \cdots, \overline{\mathcal{D}}_{t-1})$. So, we can expand $R_{\mathcal{D}_t}(\cdot)$ as a linear combination of the risks under mixture components. Note that the risk of $h_t$ under an error distribution $\overline{\mathcal{D}}_{t'}$, where $t' < t$, can equivalently be written as $R_{\overline{\mathcal{D}}_{t'}}(h_t) = \text{Pr}_{\mathcal{D}}(E_t | E_{t'})$. For $t = 0, 1, 2$ we have:

$$R_{\mathcal{D}_0}(h_0) \leq \epsilon$$

$$\frac{1}{2} R_{\mathcal{D}_0}(h_1) + \frac{1}{2} \Pr_{\mathcal{D}}(E_{h_1} | E_{h_0}) \leq \epsilon$$

$$\frac{1}{3} R_{\mathcal{D}_0}(h_2) + \frac{1}{3} \Pr_{\mathcal{D}}(E_{h_2} | E_{h_0}) + \frac{1}{3} \Pr_{\mathcal{D}}(E_{h_2} | E_{h_1}) \leq \epsilon.$$

The above constraints impose $R_{\mathcal{D}_0}(h_t) = \text{Pr}_{\mathcal{D}_0}(E_t) \leq (t+1)\epsilon$ and $\text{Pr}_{\mathcal{D}}(E_t | E_{t'}) \leq (t+1)\epsilon$. However, to limit the joint error probability of two classifiers we need to bound $\text{Pr}_{\mathcal{D}}(E_t)$:

$$\Pr_{\mathcal{D}}(E_t) = R_{\mathcal{D}}(h) \leq R_{\mathcal{D}_0}(h) + d_{\mathcal{H}\Delta\mathcal{H}}(\mathcal{D}_0, \mathcal{D}). \tag{11}$$

Finally, by a union bound, the error probability of $\text{maj}(h_0, h_1, h_2)$ is less than the sum of all pairwise joint error probabilities. Putting these all together:

$$R_{\mathcal{D}}\big(\text{maj}(h_0, h_1, h_2)\big) \leq \Pr_{\mathcal{D}}(E_{h_0} \cap E_{h_1}) + \Pr_{\mathcal{D}}(E_{h_0} \cap E_{h_2}) + \Pr_{\mathcal{D}}(E_{h_1} \cap E_{h_2})$$

$$\leq 11\epsilon^2 + 8\epsilon d_{\mathcal{H}\Delta\mathcal{H}}(\mathcal{D}_0, \mathcal{D}). \tag{12}$$

$\square$

*Proof of Theorem 3.5.* For a given domain $\mathcal{X}$, we define a new finite domain $\mathcal{X}_d \in \mathcal{X}^d$ such that $\mathcal{X}_d$ can be shattered by $\mathcal{H}$. For an arbitrary but fixed order on $\mathcal{X}_d$, we can define equivalent vector representations of functions and distributions in the new domain. Particularly, let $h : \mathcal{X} \to \{-1, 1\}$ be a binary classifier defined on $\mathcal{X}$. The vector representation of $h$ in $\mathcal{X}_d$ is $\boldsymbol{h} = (h(x))_{x \in \mathcal{X}_d} \in \{-1, 1\}^d$. Similarly, let $\mathcal{P}$ be a probability distribution over $\mathcal{X}$. The vector representation of $\mathcal{P}$ in $\mathcal{X}_d$ is $\boldsymbol{P} = (\mathcal{P}(x))_{x \in \mathcal{X}_d} \in \Delta(\mathcal{X}_d)$. Throughout the proof, we use $\mathcal{K}$ to show a subset of $\mathcal{X}_d$ with $k$ elements. We also use $\mathcal{P}(\mathcal{K})$ as a shorthand for $\sum_{x \in \mathcal{K}} \mathcal{P}(x)$. Throughout the proof we use interval notation for discrete intervals. For example, $[a, b)$ denotes $\{a, a+1, \ldots, b-1\}$. Finally, $\boldsymbol{h}_1 \circ \boldsymbol{h}_2$ denotes the entrywise product of $\boldsymbol{h}_1$ and $\boldsymbol{h}_2$.

The four allowed operations on classifiers and distributions have new interpretations in the new domain:

1. $\mathcal{D}_t \xrightarrow{\mathcal{A}} h_t$: The fact that $\mathcal{A}$ is an $\epsilon$-approximate risk minimizer imposes the following constraint:

   $$\langle \boldsymbol{h}_t \circ \boldsymbol{f}, \boldsymbol{D}_t \rangle = \sum_{h_t(x)f(x)=1} \mathcal{D}_t(x) - \sum_{h_t(x)f(x)=-1} \mathcal{D}_t(x) = 1 - 2R_{\mathcal{D}_t}(h_t) \geq 1 - 2\epsilon. \tag{13}$$

2. $h_t \xrightarrow{H} \overline{\mathcal{D}}_t$: Let $\mathcal{K}_t$ be the set of indices that $h_t$ and $f$ disagree. Then, it is straightforward to see

   $$\overline{\boldsymbol{D}}_t = \frac{1}{\mathcal{D}(\mathcal{K}_t)}(\boldsymbol{D} - \boldsymbol{h}_t \circ \boldsymbol{f} \circ \boldsymbol{D}). \tag{14}$$

3. $(h_0, h_1, \cdots, h_t) \xrightarrow{\text{maj}} \hat{h}$: This is equivalent to per dimension voting: $\hat{h}(x) = \text{maj}(h_0(x), h_1(x), \cdots, h_t(x))$ for $x \in \mathcal{X}_d$.

4. $(\mathcal{D}_0, \overline{\mathcal{D}}_0, \overline{\mathcal{D}}_1, \cdots, \overline{\mathcal{D}}_{t-1}) \xrightarrow{\text{mix}} \mathcal{D}_t$: This is simply equivalent to the weighted sum of

   $$\boldsymbol{D}_t = w_{t,0} \boldsymbol{D}_0 + \sum_{t' < t} \overline{w}_{t,t'} \overline{\boldsymbol{D}}_{t'}, \tag{15}$$

   where $w$s are the weights of the mixture components, summing up to 1.

Note that in all of the operations $h_t$ and $f$ have appeared only as $h_t \circ f$. Since $\mathcal{X}_d$ can be shattered by $\mathcal{H}$, without loss of generality, we assume $f$ is all one and use $h_t$ instead of $h_t \circ f$ in the following.

Next, we start from the initial distribution $\mathcal{D}_0$ as the input to the path dynamic benchmarks and find the constraints imposed over variables throughout the process:

- $\mathcal{D}_0 \xrightarrow{\mathcal{A}} h_0$: In this case $\langle h_0, \boldsymbol{D}_0 \rangle = 1 - 2\mathcal{D}_0(\mathcal{K}_0)$ and Equation 13 requires:

$$\mathcal{D}_0(\mathcal{K}_0) \leq \epsilon. \tag{16}$$

- $\mathcal{D}_t \xrightarrow{\mathcal{A}} h_t$: For $\boldsymbol{D}_t$ defined in Equation 15, we have:

$$\langle h_t, \boldsymbol{D}_t \rangle = w_{t,0} \langle h_t, \boldsymbol{D}_0 \rangle + \sum_{t' < t} \overline{w}_{t,t'} \langle h_t, \overline{\boldsymbol{D}}_{t'} \rangle, \tag{17}$$

where $\langle h_t, \boldsymbol{D}_0 \rangle = 1 - 2\mathcal{D}_0(\mathcal{K}_0)$ and

$$\begin{aligned}
\langle h_t, \overline{\boldsymbol{D}}_{t'} \rangle &= \frac{1}{2\mathcal{D}(\mathcal{K}_{t'})} \Big( \langle h_t, \boldsymbol{D} \rangle - \langle h_t \circ h_{t'}, \boldsymbol{D} \rangle \Big) \\
&= \frac{1}{2\mathcal{D}(\mathcal{K}_{t'})} \Big( 1 - 2\mathcal{D}(\mathcal{K}_t) - \big( 1 - 2\mathcal{D}(\mathcal{K}_t \cup \mathcal{K}_{t'} - \mathcal{K}_t \cap \mathcal{K}_{t'}) \big) \Big) \\
&= 1 - \frac{\mathcal{D}(\mathcal{K}_t \cap \mathcal{K}_{t'})}{\mathcal{D}(\mathcal{K}_{t'})}. 
\end{aligned} \tag{18}$$

Here we used $\mathcal{D}(\mathcal{K}_t \cup \mathcal{K}_{t'} - \mathcal{K}_t \cap \mathcal{K}_{t'}) = \mathcal{D}(\mathcal{K}_t) + \mathcal{D}(\mathcal{K}_{t'}) - 2\mathcal{D}(\mathcal{K}_t \cap \mathcal{K}_{t'})$. Plugging $\langle h_t, \overline{\boldsymbol{D}}_{t'} \rangle$ into Equation 17 and requiring $\langle h_t, \boldsymbol{D}_t \rangle \geq 1 - 2\epsilon$ from Equation 13, impose

$$w_{t,0} \mathcal{D}_0(\mathcal{K}_t) + \sum_{t' < t} \overline{w}_{t,t'} \frac{\mathcal{D}(\mathcal{K}_t \cap \mathcal{K}_{t'})}{\mathcal{D}(\mathcal{K}_{t'})} \leq \epsilon. \tag{19}$$

In the following, we propose $\mathcal{K}_t$s such that constraints of Equations 16 and 19 are satisfied. These are the necessary and sufficient conditions to have $h_t$s consistent with path dynamic benchmarks.

For $0 \leq t < T = \frac{2}{\epsilon}$: Consider $\mathcal{K}_t$s such that $\mathcal{K}_t \cap \mathcal{K}_{t'} = \mathcal{K}$ for any $t' < t$. Sufficient conditions to satisfy Equations 16 and 19 are

$$\mathcal{K}_t \cap \mathcal{K}_{t'} = \mathcal{K} \tag{20}$$

$$\frac{\mathcal{D}(\mathcal{K})}{\mathcal{D}(\mathcal{K}_t)} = \frac{k}{k_t} \leq \frac{\epsilon}{2} \tag{21}$$

$$\mathcal{D}_0(\mathcal{K}_t) \leq \frac{\epsilon}{2} \tag{22}$$

at every time step. Let's reorder axes in an ascending order of $\boldsymbol{D}_0$ and name them from 1 to $d$. Also assume this results in an ascending order of $\boldsymbol{D}$ as well. For example, $\mathcal{D} = \mathrm{unif}(\mathcal{X}_d)$ always satisfies this. We set $\mathcal{K} = [1, k]$ and $\mathcal{K}_0 = [1, k']$ where $k' = \frac{2k}{\epsilon}$. Then for $1 \leq t < T = \frac{2}{\epsilon}$, we set $\mathcal{K}_t = \mathcal{K} \cup (k' + (t-1)(k' - k), k' + t(k' - k)]$:

$$\mathcal{K} : \overset{k}{\longleftrightarrow}$$

$$\mathcal{K}_0 : \overset{k' = \frac{2k}{\epsilon}}{\longleftarrow\!\!\!-\!\!\!-\!\!\!-\!\!\!\longrightarrow}$$

$$\mathcal{K}_1 : \overset{k}{\longleftrightarrow} \qquad \overset{k' - k}{\longleftarrow\!\!\!-\!\!\!\longrightarrow}$$

$$\mathcal{K}_2 : \overset{k}{\longleftrightarrow} \qquad\qquad \overset{k' - k}{\longleftarrow\!\!\!-\!\!\!\longrightarrow}$$

$$\ldots$$

It is straightforward to see that this assignment satisfies Equations 20 and 21. Since axes are ordered ascendingly according to $\boldsymbol{D}_0$, a sufficient condition to satisfy Equation 22 for all rounds is

$$\mathcal{D}_0(\mathcal{K}_{T-1}) \leq \mathcal{D}_0\big(k' + (T-1)(k' - k)\big) k' \leq \mathcal{D}_0(Tk')k' = \mathcal{D}_0\Big(\frac{2k'}{\epsilon}\Big) k' \leq \frac{\epsilon}{2}. \tag{23}$$

For $y = \frac{2k'}{\epsilon}$, the above condition is $\mathcal{D}_0(y) \leq \frac{1}{y}$. Again, since axes are ordered in an ascending order of $\boldsymbol{D}_0$, $\mathcal{D}_0(y) \leq \frac{1}{d-y+1}$ (otherwise, the sum of $\boldsymbol{D}_0$ elements will go above 1). Then simple calculations show $d \geq 2y = \frac{8k}{\epsilon^2}$ is sufficient to hold Equation 23. Since $k$ is an integer, this is possible for $d \geq \frac{8}{\epsilon^2}$ which requires $\mathrm{VCdim}(\mathcal{H}) \geq \frac{8}{\epsilon^2}$. For $d = \frac{8}{\epsilon^2}$, this gives $\frac{k}{d} = \frac{\epsilon^2}{8}$.

For $t \geq T$: We set $h_t = h_{\phi(t)}$ where $\phi : [T, \infty) \to [0, T-1]$ is an assignment function working as follows. We define updated weights for $0 \leq \tau < T$:

$$v_{t,0} = w_{t,0} \tag{24}$$

$$\overline{v}_{t,\tau} = \overline{w}_{t,\tau} + \sum_{T \leq t' < t} \overline{w}_{t,t'} \mathbb{1}\{\phi(t') = \tau\}. \tag{25}$$

Then $\phi(\cdot)$ assigns round $t$ according to

$$\phi(t) = \arg\min_{0 \leq \tau < T} \overline{v}_{t,\tau}. \tag{26}$$

The sum of the updated weights satisfies

$$\sum_{0 \leq \tau < T} \overline{v}_{t,\tau} = \sum_{0 \leq \tau < T} \overline{w}_{t,\tau} + \sum_{T \leq t' < t} \overline{w}_{t,t'} \sum_{0 \leq \tau < T} \mathbb{1}\{\phi(t') = \tau\} = \sum_{0 \leq t' < t} \overline{w}_{t,t'} \leq 1, \tag{27}$$

where we used the identity $\sum_{0 \leq \tau < T} \mathbb{1}\{\phi(t') = \tau\} = 1$. This guarantees

$$\overline{v}_{t,\phi(t)} \leq \frac{1}{T} = \frac{\epsilon}{2}. \tag{28}$$

Plugging $h_t = h_{\phi(t)}$ into Equation 19 gives:

$$w_{t,0}\mathcal{D}_0(\mathcal{K}_{\phi(t)}) + \sum_{0 \leq \tau < T} \overline{w}_{t,\tau} \frac{\mathcal{D}(\mathcal{K})}{\mathcal{D}(\mathcal{K}_\tau)} + \sum_{T \leq t' < t} \overline{w}_{t,t'} \frac{\mathcal{D}(\mathcal{K})}{\mathcal{D}(\mathcal{K}_{\phi(t')})}$$

$$= w_{t,0}\mathcal{D}_0(\mathcal{K}_{\phi(t)}) + \left( \overline{w}_{t,\phi(t)} + \sum_{\substack{T \leq t' < t \\ \phi(t') = \phi(t)}} \overline{w}_{t,t'} \right) + \sum_{\substack{0 \leq \tau < T \\ \tau \neq \phi(t)}} \overline{w}_{t,\tau} \frac{\mathcal{D}(\mathcal{K})}{\mathcal{D}(\mathcal{K}_\tau)} + \sum_{\substack{T \leq t' < t \\ \phi(t') \neq \phi(t)}} \overline{w}_{t,t'} \frac{\mathcal{D}(\mathcal{K})}{\mathcal{D}(\mathcal{K}_{\phi(t')})}$$

$$\leq \left( w_{t,0} + \sum_{\substack{0 \leq \tau < T \\ \tau \neq \phi(t)}} \overline{w}_{t,\tau} + \sum_{\substack{T \leq t' < t \\ \phi(t') \neq \phi(t)}} \overline{w}_{t,t'} \right) \frac{\epsilon}{2} + \overline{v}_{t,\phi(t)} \leq \epsilon \tag{29}$$

Here we used the following observations. Since $h_\tau$ satisfies $\mathcal{D}_0(\mathcal{K}_\tau) \leq \frac{\epsilon}{2}$ for all $\tau < t$, $h_t$ also satisfies $\mathcal{D}_0(\mathcal{K}_t) = \mathcal{D}_0(\mathcal{K}_{\phi(t)}) \leq \frac{\epsilon}{2}$. Also all the ratios $\frac{\mathcal{D}(\mathcal{K})}{\mathcal{D}(\mathcal{K}_\tau)}$ and $\frac{\mathcal{D}(\mathcal{K})}{\mathcal{D}(\mathcal{K}_{\phi(t')})}$ are bounded by $\frac{\epsilon}{2}$ according to Equation 21. We finally applied Equation 28 to bound $\overline{v}_{t,\phi(t)}$. This completes our argument that $h_t$ is consistent for $t \geq T$.

So far, we have introduced a sequence of classifiers $(h_t)_t$ which are consistent with path dynamic benchmarks and all misclassify $\mathcal{K}$. We also showed for the selection of $d = \frac{8}{\epsilon^2}$, we have $\frac{k}{d} = \frac{\epsilon^2}{8}$. The only required assumption is that the element of $\boldsymbol{D}_0$ and $\boldsymbol{D}$ are both ascending. For the special case of $\mathcal{D} = \mathrm{unif}(\mathcal{X}_d)$, $\mathcal{D}(\mathcal{K}) = \frac{k}{d}$. Since elements of $\mathcal{K}$ are misclassified by all the classifiers, no matter how the majority vote of them is calculated, $\mathcal{K}$ will be in the error set:

$$R_\mathcal{D}\big(\mathrm{maj}(h_0, h_1, \cdots)\big) \geq \mathcal{D}(\mathcal{K}) = \frac{k}{d} = \frac{\epsilon^2}{8}. \tag{30}$$

This completes the theorem.

$\square$

*Proof of Theorem 3.6.* Let $\mathcal{D}_t^\delta = \mathcal{D}_t|_{x \in \mathcal{X}^\delta}$ and $\mathcal{D}_t^{\overline{\delta}} = \mathcal{D}_t|_{x \notin \mathcal{X}^\delta}$. Also let $\delta_t = \mathrm{Pr}_{\mathcal{D}_t}(x \in \mathcal{X}^\delta)$. So, we can write $\mathcal{D}_t$ as a mixture of its restricted distributions: $\mathcal{D}_t = \delta_t \mathcal{D}_t^\delta + (1-\delta_t)\mathcal{D}_t^{\overline{\delta}}$. The classifier $h_t$ trained on $\mathcal{D}_t$ should satisfy

$$R_{\mathcal{D}_t}(h_t) = \delta_t R_{\mathcal{D}_t^\delta}(h_t) + (1-\delta_t)R_{\mathcal{D}_t^{\overline{\delta}}}(h_t) = \frac{\delta_t}{2} + (1-\delta_t)R_{\mathcal{D}_t^{\overline{\delta}}}(h_t)$$

$$\leq \min_{h \in \mathcal{H}} R_{\mathcal{D}_t}(h) + \epsilon = \frac{\delta_t}{2} + \epsilon, \tag{31}$$

which requires $(1 - \delta_t)R_{\mathcal{D}_t^{\bar{\delta}}}(h_t) \leq \epsilon$. Here we used the fact that labels of $\mathcal{X}^\delta$ are equiprobable to be $1$ or $-1$. So, risk of any classifier under $\mathcal{D}_t^\delta$ is $\frac{1}{2}$. The inequality comes from $\mathcal{A}$ being $\epsilon$-approximate risk minimizer and the last equality is due the realizability of $\mathcal{D}_t^{\bar{\delta}}$. The error distribution of $h_t$ is

$$\overline{\mathcal{D}}_t = \frac{\frac{\delta}{2}\overline{\mathcal{D}}_t^\delta + (1-\delta)R_{\mathcal{D}^{\bar{\delta}}}(h_t)\overline{\mathcal{D}}_t^{\bar{\delta}}}{\frac{\delta}{2} + (1-\delta)R_{\mathcal{D}^{\bar{\delta}}}(h_t)}, \tag{32}$$

where $\mathcal{D}^{\bar{\delta}} = \mathcal{D}|_{x \notin \mathcal{X}^\delta}$, $\overline{\mathcal{D}}_t^\delta = \mathcal{D}|_{x \in \mathcal{X}^\delta, h_t(x) \neq y}$, $\overline{\mathcal{D}}_t^{\bar{\delta}} = \mathcal{D}|_{x \notin \mathcal{X}^\delta, h_t(x) \neq f(x)}$, and $\delta = \Pr_{\mathcal{D}}(x \in \mathcal{X}^\delta)$. Note that again we used the fact that labels of $\mathcal{X}^\delta$ are random. Since $\mathcal{D}_0 = \mathcal{D}$ has a weight of $\frac{1}{t+1}$ in $\mathcal{D}_t$:

$$R_{\mathcal{D}^{\bar{\delta}}}(h_t) \leq (t+1)R_{\mathcal{D}_t^{\bar{\delta}}}(h_t) \leq (t+1)\frac{\epsilon}{1-\delta_t}. \tag{33}$$

So, the weight of $\overline{\mathcal{D}}_t^\delta$ component in $\overline{\mathcal{D}}_t$ (Equation 32) will be greater than or equal to $(1 + 2(t+1)\frac{\epsilon}{\delta}\frac{1-\delta}{1-\delta_t})^{-1}$. Then the total weight given to $\mathcal{X}^\delta$ by distribution $\mathcal{D}_{t+1} = \frac{t+1}{t+2}\mathcal{D}_t + \frac{1}{t+2}\overline{\mathcal{D}}_t$ will be

$$\delta_{t+1} \geq \frac{t+1}{t+2}\delta_t + \left(\frac{1}{t+2}\right)\frac{1}{1 + 2(t+1)\frac{\epsilon}{\delta}\frac{1-\delta}{1-\delta_t}}. \tag{34}$$

The first observation from the above equation is

$$\delta_1 \geq \frac{\delta}{2} + \frac{1}{2}\frac{1}{1 + 2\frac{\epsilon}{\delta}}, \tag{35}$$

where we used $\delta_0 = \delta$. For sufficiently large $\frac{\delta}{\epsilon}$, $\delta_1$ goes above $\frac{1}{2}$ which means more than half of the benchmark will be focused on the unrealizable instances. Next, we show this is not limited to the first round, and $\delta_t$ maintains a large proportion of $\mathcal{D}_t$ as we progress. Recursively expanding Equation 34 and assuming $\delta_t \leq 2$:

$$
\begin{aligned}
\delta_t &\geq \frac{\delta}{t+1} + \frac{1}{t+1}\sum_{t'=1}^{t}\frac{1}{1 + a\,t'} \\
&\geq \frac{\delta}{t+1} + \frac{\ln(1 + a(t+1))}{a(t+1)} - \frac{\ln(1+a)}{a(t+1)} = \frac{\delta}{t+1} + \frac{\ln(1 + \frac{a\,t}{1+a})}{a(t+1)} \\
&\geq \frac{\delta}{t+1} + \frac{1}{a(t+1)}\left(\frac{at/(1+a)}{1 + at/(1+a)}\right) = \frac{\delta}{t+1} + \left(\frac{t}{t+1}\right)\frac{1}{1 + a(t+1)},
\end{aligned}
\tag{36}
$$

where $a = 4\frac{\epsilon}{\delta}(1-\delta)$. We applied $\ln(1+x) \geq \frac{x}{1+x}$ to obtain the last inequality. This shows the upperbound cannot decrease faster than $\Omega(\frac{1}{1+a\,t})$. For $t \geq 1$, we have

$$\delta_t \geq \frac{1}{2(1 + 2at)} \geq \frac{1}{2(1 + 8\frac{\epsilon}{\delta}t)}. \tag{37}$$

So, for $t \leq \beta\frac{\delta}{\epsilon}$, $\delta_t \geq \frac{1}{2(1+8\beta)}$. This completes the proof. $\square$

*Proof of Theorem 4.1.* We use the same notation as Figure 2 and follow similar arguments as the proof of Theorem 3.4. Also for notation convenience we use the shorthand $\Delta_0$ to denote $d_{\mathcal{H}\Delta\mathcal{H}}(\mathcal{D}_0, \mathcal{D})$.

Starting from the first path dynamic benchmarking step $(h_0 \to h_1 \to h_2)$, in Theorem 3.4 we observed: Now we can bound $R_{\mathcal{D}}(g_0)$ by

$$R_{\mathcal{D}}(g_0) \leq 11\epsilon^2 + 8\epsilon\Delta_0. \tag{38}$$

For the second path dynamic benchmarking step $(h_3 \to h_4 \to h_5)$ we have:

$$\frac{1}{2}R_{\mathcal{D}_0}(h_3) + \frac{1}{2}\Pr_{\mathcal{D}}(E_{h_3}|E_{g_0}) \leq \epsilon \tag{39}$$

$$\frac{1}{3}R_{\mathcal{D}_0}(h_4) + \frac{1}{3}\Pr_{\mathcal{D}}(E_{h_4}|E_{g_0}) + \frac{1}{3}\Pr_{\mathcal{D}}(E_{h_4}|E_{h_3}) \leq \epsilon \tag{40}$$

$$\frac{1}{4}R_{\mathcal{D}_0}(h_5) + \frac{1}{4}\Pr_{\mathcal{D}}(E_{h_5}|E_{g_0}) + \frac{1}{4}\Pr_{\mathcal{D}}(E_{h_5}|E_{h_3}) + \frac{1}{4}\Pr_{\mathcal{D}}(E_{h_5}|E_{h_4}) \leq \epsilon. \tag{41}$$

These equations let us bound $R_{\mathcal{D}}(g_1)$ and $\Pr_{\mathcal{D}}(E_{g_1}|E_{g_0})$:

$$R_{\mathcal{D}}(g_1) \leq \Pr_{\mathcal{D}}(E_{h_3} \cap E_{h_4}) + \Pr_{\mathcal{D}}(E_{h_3} \cap E_{h_5}) + \Pr_{\mathcal{D}}(E_{h_4} \cap E_{h_5})$$
$$\leq 3\epsilon(2\epsilon + \Delta_0) + 4\epsilon(2\epsilon + \Delta_0) + 4\epsilon(3\epsilon + \Delta_0) = 26\epsilon^2 + 11\epsilon\Delta_0, \tag{42}$$

and

$$\Pr_{\mathcal{D}}(E_{g_1}|E_{g_0}) \leq \Pr_{\mathcal{D}}(E_{h_3}|E_{g_0}) + \Pr_{\mathcal{D}}(E_{h_4}|E_{g_0}) + \Pr_{\mathcal{D}}(E_{h_5}|E_{g_0}) \leq 2\epsilon + 3\epsilon + 4\epsilon = 9\epsilon. \tag{43}$$

Finally, for the third path dynamic benchmarking step ($h_6 \to h_7 \to h_8$) we have:

$$\frac{1}{3}R_{\mathcal{D}_0}(h_6) + \frac{1}{3}\Pr_{\mathcal{D}}(E_{h_6}|E_{g_0}) + \frac{1}{3}\Pr_{\mathcal{D}}(E_{h_6}|E_{g_1}) \leq \epsilon \tag{44}$$

$$\frac{1}{4}R_{\mathcal{D}_0}(h_7) + \frac{1}{4}\Pr_{\mathcal{D}}(E_{h_7}|E_{g_0}) + \frac{1}{4}\Pr_{\mathcal{D}}(E_{h_7}|E_{g_1}) + \frac{1}{4}\Pr_{\mathcal{D}}(E_{h_7}|E_{h_6}) \leq \epsilon \tag{45}$$

$$\frac{1}{5}R_{\mathcal{D}_0}(h_8) + \frac{1}{5}\Pr_{\mathcal{D}}(E_{h_8}|E_{g_0}) + \frac{1}{5}\Pr_{\mathcal{D}}(E_{h_8}|E_{g_1}) + \frac{1}{5}\Pr_{\mathcal{D}}(E_{h_8}|E_{h_6}) + \frac{1}{5}\Pr_{\mathcal{D}}(E_{h_8}|E_{h_7}) \leq \epsilon. \tag{46}$$

This lets us bound $\Pr_{\mathcal{D}}(E_{g_2}|E_{g_0})$ and $\Pr_{\mathcal{D}}(E_{g_2}|E_{g_1})$:

$$\Pr_{\mathcal{D}}(E_{g_2}|E_{g_0}) \leq \Pr_{\mathcal{D}}(E_{h_6}|E_{g_0}) + \Pr_{\mathcal{D}}(E_{h_7}|E_{g_0}) + \Pr_{\mathcal{D}}(E_{h_8}|E_{g_0}) \leq 3\epsilon + 4\epsilon + 5\epsilon = 12\epsilon, \tag{47}$$

and a similar calculation shows $\Pr_{\mathcal{D}}(E_{g_2}|E_{g_1}) \leq 12\epsilon$. Putting these all together, we can bound the risk of $\mathrm{maj}(g_0, g_1, g_2)$ by

$$R_{\mathcal{D}}(\mathrm{maj}(g_0, g_1, g_2)) \leq \Pr_{\mathcal{D}}(E_{g_0} \cap E_{g_1}) + \Pr_{\mathcal{D}}(E_{g_0} \cap E_{g_2}) + \Pr_{\mathcal{D}}(E_{g_1} \cap E_{g_2})$$
$$\leq R_{\mathcal{D}}(g_0)\Pr_{\mathcal{D}}(E_{g_1}|E_{g_0}) + R_{\mathcal{D}}(g_0)\Pr_{\mathcal{D}}(E_{g_2}|E_{g_0}) + R_{\mathcal{D}}(g_1)\Pr_{\mathcal{D}}(E_{g_2}|E_{g_1})$$
$$\leq 9\epsilon(11\epsilon^2 + 8\Delta_0\epsilon) + 12\epsilon(11\epsilon^2 + 8\Delta_0\epsilon) + 12\epsilon(26\epsilon^2 + 11\Delta_0\epsilon)$$
$$= 543\epsilon^3 + 300\epsilon^2\Delta_0, \tag{48}$$

which completes the proof. $\qquad\square$

*Proof of Theorem 4.2.* We follow the proof of Theorem 3.5 and define equivalent vector representations of functions and distributions in a new finite domain $\mathcal{X}_d \in \mathcal{X}^d$ such that $\mathcal{X}_d$ can be shattered by $\mathcal{H}$ (so, $d \leq \frac{2}{\epsilon^3}$). We also name variables according to Figure 2. For a true classifier $f$, $\mathcal{K}_t$ denotes the set of indices where $h_t$ and $f$ disagree. Similarly, $\mathcal{K}_{g_t}$ is the set of indices where $f$ and $g_t$ disagree.

First of all, we reorder axes in an ascending order of $\boldsymbol{D}_0$. Also assume this results in an ascending order of $\boldsymbol{D}$ as well. For example, $\mathcal{D} = \mathrm{unif}(\mathcal{X}_d)$ always satisfies this. Naming axes from 1 to $d$ and

showing them on the horizontal line, we assign $\mathcal{K}_t$s and $\mathcal{K}_{g_t}$s according to:

$\mathcal{K}_0 : \longleftarrow \overset{\frac{1}{\epsilon^2}}{\longrightarrow}$

$\mathcal{K}_1 : \longleftarrow \overset{\frac{1}{\epsilon}}{\longrightarrow} \qquad \longleftarrow \overset{\frac{1}{\epsilon^2}-\frac{1}{\epsilon}}{\longrightarrow}$

$\mathcal{K}_2 : \longleftarrow \overset{\frac{1}{\epsilon}}{\longrightarrow} \qquad\qquad \longleftarrow \overset{\frac{1}{\epsilon^2}-\frac{1}{\epsilon}}{\longrightarrow}$

$\mathcal{K}_{g_0} : \longleftarrow \overset{\frac{1}{\epsilon}}{\longrightarrow}$

$\mathcal{K}_3 : \overset{1}{\longleftrightarrow} \qquad \longleftarrow \overset{\frac{1}{\epsilon^2}-1}{\longrightarrow}$

$\mathcal{K}_4 : \overset{1}{\longleftrightarrow} \quad \longleftarrow \overset{\frac{1}{\epsilon}-1}{\longrightarrow} \qquad \longleftarrow \overset{\frac{1}{\epsilon^2}-\frac{1}{\epsilon}}{\longrightarrow}$

$\mathcal{K}_5 : \overset{1}{\longleftrightarrow} \quad \longleftarrow \overset{\frac{1}{\epsilon}-1}{\longrightarrow} \qquad\qquad \longleftarrow \overset{\frac{1}{\epsilon^2}-\frac{1}{\epsilon}}{\longrightarrow}$

$\mathcal{K}_{g_1} : \overset{1}{\longleftrightarrow} \quad \longleftarrow \overset{\frac{1}{\epsilon}-1}{\longrightarrow}$

$\mathcal{K}_6 : \overset{1}{\longleftrightarrow} \qquad \longleftarrow \overset{\frac{1}{\epsilon^2}-1}{\longrightarrow}$

$\mathcal{K}_7 : \overset{1}{\longleftrightarrow} \quad \longleftarrow \overset{\frac{1}{\epsilon}-1}{\longrightarrow} \qquad \longleftarrow \overset{\frac{1}{\epsilon^2}-\frac{1}{\epsilon}}{\longrightarrow}$

$\mathcal{K}_8 : \overset{1}{\longleftrightarrow} \quad \longleftarrow \overset{\frac{1}{\epsilon}-1}{\longrightarrow} \qquad\qquad \longleftarrow \overset{\frac{1}{\epsilon^2}-\frac{1}{\epsilon}}{\longrightarrow}$

$\mathcal{K}_{g_2} : \overset{1}{\longleftrightarrow} \quad \longleftarrow \overset{\frac{1}{\epsilon}-1}{\longrightarrow}$

Next, we discuss each path dynamic benchmarking step and find sufficient conditions under which $\mathcal{K}_t$s are consistent with the dynamic benchmarking routine.

1. For the first path dynamic benchmarking step ($h_0 \to h_1 \to h_2$), let $\mathcal{K}_0 = [1, \frac{1}{\epsilon^2}]$, $\mathcal{K}_1 = [1, \frac{1}{\epsilon}] \cup (\frac{1}{\epsilon^2}, \frac{2}{\epsilon^2} - \frac{1}{\epsilon}]$, and $\mathcal{K}_2 = [1, \frac{1}{\epsilon}] \cup (2\frac{1}{\epsilon^2} - \frac{1}{\epsilon}, 3\frac{1}{\epsilon^2} - 2\frac{1}{\epsilon}]$. We have

$$\mathcal{K}_{g_0} = \mathcal{K}_0 \cap \mathcal{K}_1 = \mathcal{K}_0 \cap \mathcal{K}_2 = \mathcal{K}_1 \cap \mathcal{K}_2 = [1, \frac{1}{\epsilon}] \tag{49}$$

$$\frac{\mathcal{D}(\mathcal{K}_{g_0})}{\mathcal{D}(\mathcal{K}_1)} \leq \frac{\mathcal{D}(\mathcal{K}_{g_0})}{\mathcal{D}(\mathcal{K}_0)} \leq \epsilon. \tag{50}$$

With a similar reasoning as the proof of Theorem 3.5, the only remaining condition to ensure $h_0$, $h_1$, and $h_2$ are consistent with path dynamic benchmarks, regardless of the weightings, is $\mathcal{D}_0(\mathcal{K}_2) \leq \epsilon$. The output of this step has the error set of $\mathcal{K}_{g_0} = [1, \frac{1}{\epsilon}]$ no matter how the weighted majority vote is calculated.

2. For the second path dynamic benchmarking step ($h_3 \to h_4 \to h_5$), let $\mathcal{K}_3 = [1] \cup (\frac{1}{\epsilon}, \frac{1}{\epsilon^2} + \frac{1}{\epsilon} - 1]$, $\mathcal{K}_4 = [1] \cup (\frac{1}{\epsilon}, \frac{2}{\epsilon} - 1] \cup (\frac{1}{\epsilon^2} + \frac{1}{\epsilon} - 1, \frac{2}{\epsilon^2} - 1]$, and $\mathcal{K}_5 = [1] \cup (\frac{1}{\epsilon}, \frac{2}{\epsilon} - 1] \cup (\frac{2}{\epsilon^2} - 1, \frac{3}{\epsilon^2} - \frac{1}{\epsilon} - 1]$. We have

$$\mathcal{K}_{g_1} = \mathcal{K}_3 \cap \mathcal{K}_4 = \mathcal{K}_3 \cap \mathcal{K}_5 = \mathcal{K}_4 \cap \mathcal{K}_5 = [1] \cup (\frac{1}{\epsilon}, \frac{2}{\epsilon} - 1] \tag{51}$$

$$\frac{\mathcal{D}(\mathcal{K}_{g_1})}{\mathcal{D}(\mathcal{K}_4)} \leq \frac{\mathcal{D}(\mathcal{K}_{g_1})}{\mathcal{D}(\mathcal{K}_3)} \leq \epsilon \tag{52}$$

$$\frac{\mathcal{D}(\mathcal{K}_3)}{\mathcal{D}(\mathcal{K}_{g_0})} = \frac{\mathcal{D}(\mathcal{K}_4)}{\mathcal{D}(\mathcal{K}_{g_0})} = \frac{\mathcal{D}(\mathcal{K}_5)}{\mathcal{D}(\mathcal{K}_{g_0})} \leq \epsilon. \tag{53}$$

It is easy to check that for any weighting of mixture components, if $\mathcal{D}_0(\mathcal{K}_5) \leq \epsilon$, $h_3$, $h_4$, and $h_5$ will be consistent with path dynamic benchmarks and the output of this step has the error set of $\mathcal{K}_{g_1} = [1] \cup (\frac{1}{\epsilon}, \frac{2}{\epsilon} - 1]$. Note that $\mathcal{D}_0(\mathcal{K}_5) \leq \epsilon$ guarantees $\mathcal{D}_0(\mathcal{K}_2)$ as well.

3. For the third path dynamic benchmarking step ($h_6 \rightarrow h_7 \rightarrow h_8$), let $\mathcal{K}_6 = [1] \cup (\frac{2}{\epsilon} - 1, \frac{1}{\epsilon^2} + \frac{2}{\epsilon} - 2]$, $\mathcal{K}_7 = [1] \cup (\frac{2}{\epsilon} - 1, \frac{3}{\epsilon} - 2] \cup (\frac{1}{\epsilon^2} + \frac{2}{\epsilon} - 2, \frac{2}{\epsilon^2} + \frac{1}{\epsilon} - 2]$, and $\mathcal{K}_8 = [1] \cup (\frac{2}{\epsilon} - 1, \frac{3}{\epsilon} - 2] \cup (\frac{2}{\epsilon^2} + \frac{1}{\epsilon} - 2, \frac{3}{\epsilon^2} - 2]$. We have

$$\mathcal{K}_{g_2} = \mathcal{K}_6 \cap \mathcal{K}_7 = \mathcal{K}_6 \cap \mathcal{K}_8 = \mathcal{K}_7 \cap \mathcal{K}_8 = [1] \cup \left(\frac{2}{\epsilon} - 1, \frac{3}{\epsilon} - 2\right] \tag{54}$$

$$\frac{\mathcal{D}(\mathcal{K}_{g_2})}{\mathcal{D}(\mathcal{K}_7)} \leq \frac{\mathcal{D}(\mathcal{K}_{g_2})}{\mathcal{D}(\mathcal{K}_6)} \leq \epsilon \tag{55}$$

$$\frac{\mathcal{D}(\mathcal{K}_6)}{\mathcal{D}(\mathcal{K}_{g_0})} = \frac{\mathcal{D}(\mathcal{K}_7)}{\mathcal{D}(\mathcal{K}_{g_0})} = \frac{\mathcal{D}(\mathcal{K}_8)}{\mathcal{D}(\mathcal{K}_{g_0})} \leq \epsilon \tag{56}$$

$$\frac{\mathcal{D}(\mathcal{K}_6)}{\mathcal{D}(\mathcal{K}_{g_1})} = \frac{\mathcal{D}(\mathcal{K}_7)}{\mathcal{D}(\mathcal{K}_{g_1})} = \frac{\mathcal{D}(\mathcal{K}_8)}{\mathcal{D}(\mathcal{K}_{g_1})} \leq \epsilon. \tag{57}$$

Again, it is straightforward to see that for any weighting, if $\mathcal{D}_0(\mathcal{K}_8) \leq \epsilon$, $h_6$, $h_6$, and $h_8$ will be consistent with path dynamic benchmarks and the output of this step has the error set of $\mathcal{K}_{g_2} = [1] \cup (\frac{2}{\epsilon} - 1, \frac{3}{\epsilon} - 2]$. Note that $\mathcal{D}_0(\mathcal{K}_8) \leq \epsilon$ guarantees $\mathcal{D}_0(\mathcal{K}_5)$ as well.

Putting these all together, to make the provided classifiers consistent with hierarchical dynamic benchmarking, it suffices to show $\mathcal{D}_0(\mathcal{K}_8) \leq \epsilon$:

$$\mathcal{D}_0(\mathcal{K}_8) \leq \mathcal{D}_0\left(\frac{3}{\epsilon^2} - 2\right)\left(\frac{1}{\epsilon^2} - \frac{1}{\epsilon}\right) \leq \epsilon. \tag{58}$$

Since axes are ordered in an ascending order of $\boldsymbol{D}_0$, $\mathcal{D}_0(x) \leq \frac{1}{d-x+1}$. Otherwise, the sum of the $\boldsymbol{D}_0$ elements would go above 1. So, we have:

$$\frac{1}{d - (\frac{3}{\epsilon^2}) + 1}\left(\frac{1}{\epsilon^2} - \frac{1}{\epsilon}\right) \leq \epsilon \tag{59}$$

which imposes $d \geq \frac{1}{\epsilon^3} + \frac{2}{\epsilon^2} - 1$. For any valid $\epsilon$, i.e. $\epsilon \geq \frac{1}{2}$, $d = \frac{2}{\epsilon^3}$ satisfies this condition. It also satisfies the other limitation we had on $\text{VCdim}(\mathcal{H})$. In this case

$$R_{\mathcal{D}}\big(\text{maj}(g_0, g_1, g_2)\big) \geq \mathcal{D}(\mathcal{K}_{g_0} \cap \mathcal{K}_{g_1} \cap \mathcal{K}_{g_2}) = \mathcal{D}([1]). \tag{60}$$

For the choice of $\mathcal{D} = \text{unif}(\mathcal{X}_d)$, this risk is $\frac{1}{d} = \frac{\epsilon^3}{2}$ which completes the proof.

$\square$

*Proof of Lemma A.1.* The derivative of $R_{\mathcal{D}}^{\text{hinge}}(h)$ w.r.t $h(x)$ is:

$$g(x) := \frac{\partial R_{\mathcal{D}}^{\text{hinge}}}{\partial h(x)} = \begin{cases} 0 & h(x)f(x) \geq 1 \\ -f(x)\mathcal{D}(x) & o.w. \end{cases}. \tag{61}$$

We can write $\boldsymbol{g}$ with vector representations as

$$\boldsymbol{g} = -\Pr_{\mathcal{D}}(E_h)\,(\boldsymbol{f} \circ \overline{\boldsymbol{D}}_h) = -R_{\mathcal{D}}^{\text{hinge}}(h)\,(\boldsymbol{f} \circ \overline{\boldsymbol{D}}_h). \tag{62}$$

As mentioned earlier, updates of the form $\boldsymbol{h} := \boldsymbol{h} - \eta\boldsymbol{g}$ are infeasible as in practice $\overline{\boldsymbol{D}}_h$ is only available at limited samples. To make it practical, let $\overline{h} = \mathcal{A}(\overline{\mathcal{D}}_h)$. As $\mathcal{A}$ is an $\epsilon$-approximate risk minimizer,

$$R_{\overline{\mathcal{D}}_h}(\overline{h}) = \frac{1}{2}\langle \boldsymbol{1} - \overline{\boldsymbol{h}} \circ \boldsymbol{f}, \overline{\boldsymbol{D}}_h \rangle = \frac{1}{2} - \frac{1}{2}\langle \overline{\boldsymbol{h}}, \boldsymbol{f} \circ \overline{\boldsymbol{D}}_h \rangle \leq \epsilon. \tag{63}$$

So, we have $\langle \overline{\boldsymbol{h}}, \boldsymbol{f} \circ \overline{\boldsymbol{D}}_h \rangle \geq 1 - 2\epsilon$. For $\epsilon < 0.5$, $\overline{\boldsymbol{h}}$ will be a descent direction. $\square$

*Proof of Lemma A.2.* The derivative of $R_{\mathcal{D}}^{\exp}(h)$ w.r.t $h(x)$ is

$$g(x) := \frac{\partial R_{\mathcal{D}}^{\exp}}{\partial h(x)} = -f(x)\mathcal{D}(x)\exp(-h(x)f(x)) \tag{64}$$

or in the vector space

$$\boldsymbol{g} = -\boldsymbol{f} \circ \boldsymbol{D} \circ \exp(-\boldsymbol{h} \circ \boldsymbol{f}). \tag{65}$$

To find a good descent direction, we solve for

$$\min_{\tilde{\boldsymbol{h}} \in \mathcal{H}} \langle \tilde{\boldsymbol{h}}, \boldsymbol{g} \rangle = \langle -\tilde{\boldsymbol{h}} \circ \boldsymbol{f}, \boldsymbol{D} \circ \exp(-\boldsymbol{h} \circ \boldsymbol{f}) \rangle. \tag{66}$$

The solution to this problem is the minimizer of $R_{\mathcal{D}_h}(\tilde{h})$ where $\mathcal{D}_h$ is the true distribution weighted according to $\mathcal{D}_h(x) = \frac{1}{Z_h}\mathcal{D}(x)\exp(-h(x)f(x))$ and $Z_h$ is a normalization factor to make sure $\mathcal{D}_h$ will be a probability distribution. Let $\tilde{h} = \mathcal{A}(\mathcal{D}_h)$. Given $\mathcal{A}$ is $\epsilon$-approximate risk minimizer,

$$R_{\mathcal{D}_h}(\tilde{h}) = \frac{1}{2}\langle \boldsymbol{1} - \tilde{\boldsymbol{h}} \circ \boldsymbol{f}, \boldsymbol{D}_h \rangle = \frac{1}{2} - \frac{1}{2Z_h}\langle \tilde{\boldsymbol{h}} \circ \boldsymbol{f}, \boldsymbol{D} \circ \exp(-\boldsymbol{h} \circ \boldsymbol{f}) \rangle \leq \epsilon. \tag{67}$$

So, we have $\langle \tilde{\boldsymbol{h}} \circ \boldsymbol{f}, \boldsymbol{D} \circ \exp(-\boldsymbol{h} \circ \boldsymbol{f}) \rangle \geq Z_h(1 - 2\epsilon)$. For $\epsilon < 0.5$, $\tilde{h}$ will be a descent direction and the updating rule is

$$\boldsymbol{h} := \boldsymbol{h} + \eta \tilde{\boldsymbol{h}}. \tag{68}$$

Finally, in order to find the optimum step size $\eta$, we plug the updated $\boldsymbol{h}$ into Equation 8 and solve for

$$\min_{\eta} R_{\mathcal{D}}(h + \eta \tilde{h}) = \langle \exp(-(\boldsymbol{h} + \eta \tilde{\boldsymbol{h}}) \circ \boldsymbol{f}), \boldsymbol{D} \rangle. \tag{69}$$

Since the objective is a convex function of $\eta$, it suffices to check the first order condition:

$$\begin{aligned}
\frac{\partial R_{\mathcal{D}}(h + \eta \tilde{h})}{\partial \eta} &= \langle \tilde{\boldsymbol{h}} \circ \boldsymbol{f} \circ \exp(-(\boldsymbol{h} + \eta \tilde{\boldsymbol{h}}) \circ \boldsymbol{f}), \boldsymbol{D} \rangle \\
&= \langle \tilde{\boldsymbol{h}} \circ \boldsymbol{f} \circ \exp(-\eta \tilde{\boldsymbol{h}} \circ \boldsymbol{f}), \boldsymbol{D} \circ \exp(-\boldsymbol{h} \circ \boldsymbol{f}) \rangle \\
&= Z_h \exp(-\eta)(1 - R_{\mathcal{D}_h}(\tilde{h})) - Z_h \exp(\eta)R_{\mathcal{D}_h}(\tilde{h}) = 0. \tag{70}
\end{aligned}$$

Solving the last equation gives

$$\eta = \frac{1}{2}\log\left(\frac{1}{R_{\mathcal{D}_h}(\tilde{h})} - 1\right). \tag{71}$$

$\square$

**Theorem C.2.** *For any $\epsilon$-approximate risk minimizer $\mathcal{A}$ with $\frac{1}{\epsilon} \in \mathbb{N}$, and path dynamic benchmark with $L \geq 3$ rounds of model building and arbitrary mixture weights, there exists a hypothesis class $\mathcal{H}$ with $\mathrm{VCdim}(\mathcal{H}) \leq 4$ such that for any true classifier $f \in \mathcal{H}$, there exists an underlying distribution $\mathcal{D}$ where for any initial distribution $\mathcal{D}_0$ with $\mathrm{supp}(\mathcal{D}_0) \subseteq \mathrm{supp}(\mathcal{D})$ that satisfies $\mathcal{D}(x_2) \geq \mathcal{D}(x_1) \iff \mathcal{D}_0(x_2) \geq \mathcal{D}_0(x_1)$, there exists a sequence $(h_t)_{t=0}^{T-1}$ of classifiers consistent with the path dynamic benchmark where a similar lower-bound as Theorem 3.5 holds.*

*Proof.* The proof is straightforward given the results obtained so far in the proof of Theorem 3.5. The way we constructed $\mathcal{K}_t$s is equivalent to selecting two subsets of $\mathbb{R}^+$ and changing their predicted labels. So, let $\mathcal{H}'$ be the class of such functions, i.e., the class of unions of two intervals ($\mathrm{VCdim}(\mathcal{H}') = 4$). For any $f \in \mathcal{H}'$, we select $d$ points from the real line such that $f(x)$ is all 1 or all $-1$. Let $\mathcal{X}_d \in \mathbb{R}^d$ be the set of the selected points. Then we induce a probability distribution $\mathcal{D}$ on $\mathcal{X}_d$ such that $\mathcal{D}(x) = \frac{1}{d} + \alpha(x)$ where $\alpha(x)$ is an ascending function of $x$ and $\sum_{x \in \mathcal{X}_d} \alpha(x) = 0$. Then all of our arguments in the first part of the proof will be true for any $\mathcal{D}_0$ which is ascending or descending w.r.t $x$. In the limit we have

$$\lim_{\alpha \to 0} R_{\mathcal{D}}\big(\mathrm{maj}(h_0, \cdots, h_{L-1})\big) \geq \lim_{\alpha \to 0} \mathcal{D}(\mathcal{K}) = \frac{\epsilon^2}{8}. \tag{72}$$

This completes the proof. $\square$

**Theorem C.3.** *For any $\epsilon$-approximate risk minimizer $\mathcal{A}$ with $\frac{1}{\epsilon} \in \mathbb{N}$, and hierarchical dynamic benchmark with depth-2 and width-3 (Figure 2) with arbitrary mixture and majority weights, there exists a hypothesis class $\mathcal{H}$ with $\mathrm{VCdim}(\mathcal{H}) \leq 6$ such that for any true classifier $f \in \mathcal{H}$, there exists an underlying distribution $\mathcal{D}$ where for any initial distribution $\mathcal{D}_0$ with $\mathrm{supp}(\mathcal{D}_0) \subseteq \mathrm{supp}(\mathcal{D})$ that satisfies $\mathcal{D}(x_2) \geq \mathcal{D}(x_1) \iff \mathcal{D}_0(x_2) \geq \mathcal{D}_0(x_1)$, there exists classifiers consistent with the hierarchical dynamic benchmark for which a similar lower-bound as Theorem 4.2 holds.*

*Proof.* Given the results obtained in the proof of Theorem 4.2, the proof of this theorem is pretty straightforward. The way we chose $\mathcal{K}_t$s in the proof of Theorem 4.2 is equivalent to choosing three intervals from the real line. So, let $\mathcal{H}'$ be such class of function ($\mathrm{VCdim}(\mathcal{H}') = 6$). For any $f \in \mathcal{H}'$, we select $d$ points from the real line such that $f(x)$ is all 1 or all $-1$. Let $\mathcal{X}_d \in \mathbb{R}^d$ be the set of the selected points. Then we induce a probability distribution $\mathcal{D}$ on $\mathcal{X}_d$ such that $\mathcal{D}(x) = \frac{1}{d} + \alpha(x)$ where $\alpha(x)$ is an ascending function of $x$ and $\sum_{x \in \mathcal{X}_d} \alpha(x) = 0$. All of our arguments in the first part of the proof hold for any $\mathcal{D}_0$ which is ascending or descending w.r.t $x$. In the limit we have

$$\lim_{\alpha \to 0} R_{\mathcal{D}}(\mathrm{maj}(g_0, g_1, g_2)) \geq \lim_{\alpha \to 0} \mathcal{D}([1]) = \frac{\epsilon^3}{2}, \tag{73}$$

which completes the proof. $\qquad\square$

