# OpenReview forum: "A Theory of Dynamic Benchmarks"
_ICLR.cc/2023/Conference — ICLR 2023 poster_

### Official Review · Reviewer_8sGC · 2022-10-22

**Confidence:** 3
**Correctness:** 4
**Technical Novelty And Significance:** 4
**Empirical Novelty And Significance:** 4
**Recommendation:** 8

**Clarity, Quality, Novelty And Reproducibility:**

The paper is quite clear and the results are very well presented and explained.

**Strength And Weaknesses:**

The paper first introduces the natural and intuitive path dynamic benchmarking architecture and provides the following results: in the realizable setting under the proposed  sequential architecture, it is shown that the trained model will get $O(\epsilon^2)$ error in 3 rounds and there is an $\Omega(\epsilon^2)$ lower bound for any larger number of rounds. Namely, there is no provable benefit to dynamic data collection beyond 3 rounds in the path model. The authors also show some negative results when one has label noise.

Moreover, since the above sequential model has provable limitations, it is asked whether there exist different ways to design dynamic benchmarking architectures. In particular, a “hierarchical” model is proposed that achieves error $O(\epsilon^3)$ when used with depth 2.

I find this paper extremely interesting. I really enjoyed reading the paper and I believe that the result constitutes an important contribution to learning theory. At a technical level, the lower bound for the path dynamic model is very nice and perhaps surprising. The paper is clearly written and the problem and results are clearly organized.

I would like to conclude with some questions for the authors:

Q1. In Theorem 3.5, the statement says that there exists a sequence of classifiers that is consistent but whose weighted majority has high error. Is it possible to lower bound the probability of observing such a bad sequence?

Q2. Is it possible to extend the negative result for any Boolean function (in particular, Boolean functions that could potentially satisfy a variant of Proposition 3.2)?


**Summary Of The Paper:**

This paper theoretically studies dynamic benchmarking. Given that static datasets raise various concerns when used as ML benchmarks, dynamic benchmarks are proposed by researchers. In this setting, the ML model is trained against the current dataset, while the next dataset will contain new data points chosen to challenge previously built models. The main question is whether one can design such dynamic benchmarks so that the model's performance will improve as more and more difficult instances are added to the dataset. The authors provide a theoretical model for this problem and prove some negative and positive results, which are presented in the next section.


**Summary Of The Review:**

This paper adopts a theoretical viewpoint regarding dynamic benchmarking and provides interesting lower and upper bounds for the proposed theoretical models, which are natural. I found this paper very interesting and I vote for acceptance.

---

> ### Author Response · Authors · 2022-11-15
> **Thank you!**
>
> Thank you for the detailed and complete review of our paper! We’re excited that you found our study interesting and an important contribution.
>
> Regarding Q1 and the probability of observing a bad sequence, we would like to mention that this is a great question. Our theoretical framework shows that a bad sequence exists, and we, too, were curious how likely bad sequences are in practice. We therefore studied this important question empirically in Section 5, where we 1) proposed a concrete way to simulate the performance of a dynamic benchmark, and 2) simulated path dynamic benchmarks on popular datasets with typical models and learning procedures. As our empirical results suggest, bad sequences are likely to be encountered in practice, making the performance stall long before achieving the best possible risk.
>
> Regarding Q2 and the possibility of extending negative results for any Boolean function, we think this is a great question for further study. For now, we have limited our model combination to weighted majority voting as it is a natural and stable choice.

---

### Official Review · Reviewer_nUdN · 2022-10-25

**Confidence:** 3
**Correctness:** 3
**Technical Novelty And Significance:** 3
**Empirical Novelty And Significance:** 3
**Recommendation:** 6

**Clarity, Quality, Novelty And Reproducibility:**

- Clarity: the paper is generally well-structured, but some parts are still unclear to me (please see the first point of weakness for more details).
- Quality: the theorems seem correct. But I am confused about in which case the annotators can access $\mathcal{D}$ and why they do not just collect the data from $\mathcal{D}$ at once.
- Novelty: the notion of the dynamic benchmark is novel to me.
- Reproducibility: proofs of the theorems are provided.

**Strength And Weaknesses:**

The strengths of this paper are as follows:
- the notion of dynamic benchmarks is novel to me.
- this paper shows the limitation of the popular path dynamic benchmark, which is interesting and instructive.

My concerns about the paper are as follows:

- About the readability: I am a little bit concerned about the readability of this paper, particularly about the insufficient background introduction. As mentioned in the introduction, the dynamic benchmark is proposed to address the limitation of the static benchmark. But, a clear definition of the static benchmark and the motivation to study it is not provided in the paper. The authors have provided some references, but I think it would be better to at least include some basic definitions to make the paper self-containedness.

- About the ability of the annotator: this paper assumes that annotators can access the conditional density of the misclassified instance over the unknown underlying distribution $\mathcal{D}$. When is this assumption satisfied? If $\mathcal{D}$ is accessible, why do the annotators not just collect the data at once? It would be nice if the authors could provide some practical examples.

- About the related work: I think It would be nice to provide some discussion on the difference between active learning since both work aim to improve the data collection efficiency to improve the model's performance on a fixed unknown distribution.

- Is there any theoretical result for the general depth-$k$ structure? Can this hierarchical structure continuously improve performance with deeper layers or stall with a certain depth?


**Summary Of The Paper:**

This paper studies the benefits and limitations of dynamic benchmarking from a theoretical view. For the model where data collection and model fitting alternate sequentially, the authors show the model performance improves initially but can stall after only three rounds for the commonly used path dynamic benchmarking. The authors propose a new model with a hierarchical dependency structure to circumvent this issue. The authors prove the proposed model enjoys a better theoretical guarantee than the former one, albeit with a significant increase in complexity. The empirical results support the theoretical analysis well.

**Summary Of The Review:**

Overall, the notion of dynamic benchmarking is novel to me and the proposed theorems on path dynamic benchmarking and hierarchical benchmarking are interesting. But, I am still a little bit confused about the difference between this line of research and other research fields for improving data collection efficiency (like active learning). Besides, the ability of the annotator to access the unknown distribution $\mathcal{D}$ is not clearly justified. I will raise my score if the author could address the above concerns.

---

> ### Author Response · Authors · 2022-11-15
> **Thank you! Access to D is clarified. Connection to related works is explained.**
>
> We’re encouraged that you found our study novel, interesting, and instructive. We thank you for the comments and clarifying questions and will address them in the following.
>
> Regarding the access to $\mathcal{D}$, it is correct that the data annotators can access the target distribution $\mathcal{D}$. However, by our assumption, the model builders can only find an $\epsilon$-risk minimizer on this distribution. The goal of our problem formulation is to achieve an error much smaller than $\epsilon$ through a dynamic benchmark design. This cannot be done by training on $\mathcal{D}$ directly. The motivation for this problem formulation is that the distribution $\mathcal{D}$ might have positive weight on all the examples we’re interested in. However, an error set of measure $\epsilon$ might contain many important instances that we’d like to eventually classify correctly by running a dynamic benchmark.
>
> Regarding the background introduction, we agree that providing further discussion of static benchmarks and concerns around them will make the paper more self-contained, and thank you for this suggestion.
>
> Regarding related works, we agree that further distinction of dynamic benchmarking with related techniques like active learning or boosting can be insightful and will include further discussion around it in the final manuscript. In short, an active learner chooses a sample and asks the annotators to label it. However, in dynamic benchmarks, annotators provide adversarial examples that fool a given model. In contrast with active learning, 1) we do not restrict annotators to specific instances, and 2) annotators provide both the instance and its label, as we might not even have good examples to ask the annotators to label it. It’s worth mentioning we have also included a natural connection to a complementary method inspired by boosting technique in Section A of the appendix.
>
> Finally, regarding the possibility of having theoretical results for depth-$k$ hierarchical design: We have a positive theoretical result on depth-$k$ structures, which shows a strict separation in terms of the dependence on $\epsilon$. However, a depth-$k$ width-$w$ design requires $w^k$ rounds of data collection. No dynamic benchmark has ever had more than 20 rounds. So, even though theoretically interesting, we found such complex designs to limited applicability.

---

> > ### Comment · Reviewer_nUdN · 2022-11-19
> > **discussion**
> >
> > Thank you for the feedback. It is now clear to me on the issue of the accessibility of the distribution $\mathcal{D}$. I think it would be nice to incorporate further discussion on the related work including active learning or boosting in the next version. I would like to raise my score to a positive evaluation.

---

### Official Review · Reviewer_SkQx · 2022-10-30

**Confidence:** 2
**Correctness:** 3
**Technical Novelty And Significance:** 3
**Empirical Novelty And Significance:** 3
**Recommendation:** 6

**Clarity, Quality, Novelty And Reproducibility:**

The paper talks about interesting and important topics related to  dynamic benchmarks with good quality and clarity. The idea is novel as far as I know.

**Strength And Weaknesses:**


Strength: This paper is well written and the topic of dynamic benchmarking is important and interesting. The theoretical results are sound and illuminate the benefits and practical limitations of dynamic benchmarking.

Weakness: The proposed hierarchical structure alternative is with limited applicability (due to huge computational complexity).


**Summary Of The Paper:**

This paper initiates a theoretical study of dynamic benchmarking. It is proved that model performance could stall with alternative data collection and model fitting. Therefore, a new model where data collection and model fitting have a hierarchical dependency structure is proposed and proved a better progress but with much larger complexity. Some simulations are provided to support the theoretical findings.


**Summary Of The Review:**

Overall, I think that the studied model in this paper is well-motivated. The idea behind the hierarchical structure is novel despite the complexity issues. Therefore, I recommend “weak accept”.

---

> ### Author Response · Authors · 2022-11-15
> **Thank you!**
>
> We’re glad that you found our study interesting, important, and clear. We do agree that hierarchical benchmarks may be difficult to implement in practice. This points at the difficulty of achieving a dynamic benchmark design that goes beyond the basic one, which we show has serious limitations. Our positive results on hierarchical designs are best thought of as a proof of concept showing that, in principle, there remains some hope that more sophisticated designs can avoid the impossibility results that hold for the basic path design.

---

> > ### Comment · Reviewer_SkQx · 2022-12-12
> > **Thanks for the response**
> >
> > Sorry for the late reply. After carefully reading the response, I will keep my initial rating due to the huge computational complexity.

---

### Decision · Program_Chairs · 2023-01-20

**Decision:**

Accept: poster

**Justification For Why Not Higher Score:**

It is no immediately obvious what the practical implications of these theory are in the short term.

**Justification For Why Not Lower Score:**

The paper provides a rigorous formulation of ways by which the problem can be solved (including what it means to solve it), leading to the introduction to new ways of looking at the problem via the concept of hierarchical dynamic benchmarks.

**Metareview: Summary, Strengths And Weaknesses:**

This is a theoretical paper of general interest: collecting datasets to challenge models provides up to now for a given task. Although this basic idea of dynamic benchmarking may sound appealing as a way of avoiding to label unnecessary points, many pitfalls exist, including just drifting away from covering typical but originally easier cases.

Strengths: a rigorous formulation of ways by which the problem can be solved (including what it means to solve it), leading to the introduction to new ways of looking at the problem via the concept of hierarchical dynamic benchmarks.

Weaknesses: it is no immediately obvious what the practical implications of these theory are in the short term.


**Note From Pc:**

if the above contains the word "oral" or "spotlight" please see: "oral" presentation means -> notable-top-5% and "spotlight" means -> notable-top-25%. As stated in our emails, we are disassociating presentation type from AC recommendations